# *HyPoGen*: Optimization-Biased Hypernetworks for Generalizable Policy Generation

**Hanxiang Ren**[*†]  **Li Sun**[*]  **Xulong Wang**  **Pei Zhou**  **Zewen Wu**
ZJU & HKU   HKU    ZJU     HKU  HKU & Transcengram

**Siyan Dong**  **Difan Zou**  **Youyi Zheng**[‡]  **Yanchao Yang**[‡ §]
 HKU     HKU    ZJU      HKU

## Abstract

Policy learning through behavior cloning poses significant challenges, particularly when demonstration data is limited. In this work, we present *HyPoGen*, a novel optimization-biased hypernetwork for policy generation. The proposed hypernetwork learns to synthesize *optimal policy parameters solely from task specifications* – without accessing training data – by modeling policy generation as an approximation of the optimization process executed over a finite number of steps and assuming these specifications serve as a sufficient representation of the demonstration data. By incorporating structural designs that bias the hypernetwork towards optimization, we can improve its generalization capability while only training on source task demonstrations. During the feed-forward prediction pass, the hypernetwork effectively performs an optimization in the latent (compressed) policy space, which is then decoded into policy parameters for action prediction. Experimental results on locomotion and manipulation benchmarks show that *HyPoGen* significantly outperforms state-of-the-art methods in generating policies for unseen target tasks without any demonstrations, achieving higher success rates and underscoring the potential of optimization-biased hypernetworks in advancing generalizable policy generation. Our code and data are available at: https://github.com/ReNginx/HyPoGen.

## 1 Introduction

Behavior cloning (BC) (Pomerleau, 1991) is highly promising in real-world applications, as it trains agents directly from expert demonstrations, which can be more easily acquired in certain scenarios than designing a proper reward function for Reinforcement Learning (RL). Moreover, BC alleviates the complexity in training for long-horizon tasks that may exist in infinite-dimensional spaces.

While BC can bypass the challenges of reward function design and RL training difficulties by learning directly from expert demonstrations, it comes with its own limitations: (i) *Substantial data requirement*. Gathering sufficient high-quality expert data is time-consuming and labor-intensive, as it requires days, if not months, of manual recording of numerous instances of the desired behavior (spatial-temporal trajectories). This process can be particularly burdensome in complex or hazardous tasks. (ii) *Insufficient generalization ability*. Policies trained in one environment often fail to perform well in different ones due to overfitting to specific scene variations and task dynamics encountered during training (Zhang et al., 2018; Song et al., 2020). Such a lack of generalization limits the policy's applicability in complex, real-world scenarios with varying conditions.

In response to these challenges, recent work attempts to enhance the generalization capability through Hypernetworks (Ha et al., 2017; Rezaei-Shoshtari et al., 2023). These networks learn a mapping from the task embedding (specification) space to the parameter space of target (policy) networks, enabling capturing the similarity between different tasks, which is crucial for transferability, and promoting better utilization of limited demonstrations.

---

[*]: First Authors (hanxiang.ren@zju.edu.cn, sunlids@connect.hku.hk)
[†]: Work done as a Research Assistant at the InfoBodied AI Lab at HKU
[‡]: Corresponding Authors
[§]: Senior Author

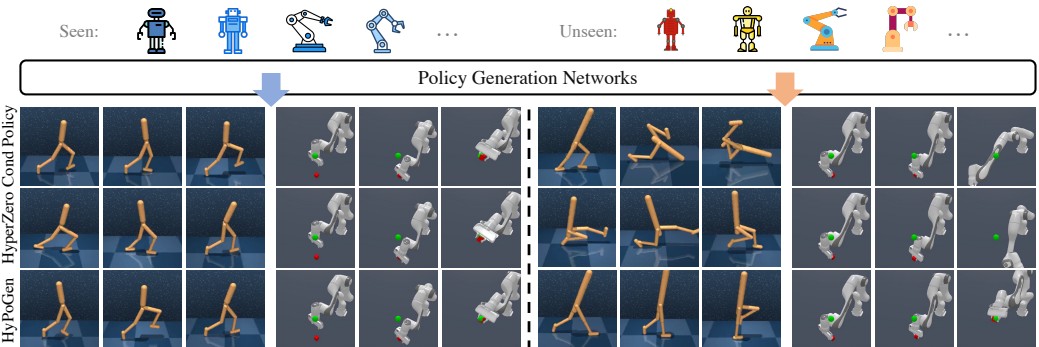

Figure 1: Policy generation networks are trained on source (*seen*) task specifications with corresponding demonstrations using behavior cloning, and then directly applied to target (*unseen*) task specifications to generate policy network parameters for action prediction. By introducing optimization inductive biases into the proposed hypernetwork *HyPoGen*, we can achieve better performance on out-of-domain specifications without resorting to additional demonstration collection and training.

Despite their success, previous works usually employ hypernetworks in a target-network-agnostic manner, typically by structuring them as multi-layer perceptrons (MLPs), and are subject to several limitations: (i) ***Overfitting risk***: Training a hypernetwork to map a task embedding to the set of parameters of a policy network faces the same problem of overfitting to the tasks being learned, similar to the generalization difficulty encountered in supervised learning. (ii) ***Disregard for the optimization process***: These hypernetworks do not account for the fact that the optimal weights of a target network are the result of optimization through (stochastic) gradient descent. (iii) ***Ignorance of target network structure***: The hypernetworks predict different parts of the policy network simultaneously and do not utilize the information flow induced by the structure of the target network.

These issues hinder the deployment of hypernetworks at a large scale for generalizable policy synthesis, for which we attempt to investigate and propose a potentially in-principle solution. Generalization from completely unrelated tasks, like piano playing to pizza cooking, is exceptionally challenging. A more achievable type of generalization occurs in tasks with similar characteristics, such as an agent learning to handle objects of various shapes or operate under unseen dynamics. In such cases, tasks are defined by environmental and embodiment parameters (specification) that help the agent leverage and generalize through sharable characteristics across different tasks.

To address these issues, we present **HyPoGen**, a method that biases **Hy**pernetworks for **Po**licy **Gen**eration from task specifications. HyPoGen employs inductive biases in both the optimization process and the interdependencies within the target network structure to facilitate generalizable policy synthesis. Our approach imitates the iterative optimization in the latent parameter space, updating policy network parameters *solely* based on task specifications rather than relying on *manually* collected demonstration data. To further bias the hypernetwork training towards optimization with structural information of the policy network, HyPoGen mimics the interdependent gradient computation (involved in the backward propagation) during the (forward) synthesis procedure in the latent parameter space. This is achieved by predicting the neural "gradients" of each conceptual network block according to the chain rule relationship. The predicted latent parameters can then be decoded back to the original parameter space, enabling action prediction when loaded into the target policy network.

We conduct an extensive evaluation within both locomotion and manipulation environments. Our proposed method, trained on the DeepMind Control Suite (Tunyasuvunakool et al., 2020) and ManiSkill (Mu et al., 2021), outperforms state-of-the-art (SOTA) methods in both settings. HyPoGen achieves a significant enhancement in success rate for various stiffness and arm lengths in ManiSkill environments. Furthermore, our experiments validate that HyPoGen performs optimization based solely on task specifications without actual demonstration data. In summary, our main contributions include:

- We propose a novel hypernetwork architecture termed *HyPoGen* that integrates the inductive bias of the optimization process of the target network under behavior cloning.
- We introduce an effective approach for learning (latent) neural gradient updates, explicitly modeling the interdependencies of gradients across different target network blocks.
- We comprehensively benchmark our approach against previous methods in both locomotion and manipulation tasks, demonstrating its superiority in generalizable policy generation.

## 2 RELATED WORK

**Behavior cloning and policy generation** BC (Bain & Sammut, 1995) enables efficient learning and replication of policies through expert demonstration data. Due to its ability to directly imitate demonstrated behavior without requiring fine-grained reward functions or extensive interactions with the environment, BC has shown remarkable effectiveness when combined with RL tasks (Lee & Zhang, 2021; Goecks et al., 2019; Kumar et al., 2022; Rajeswaran et al., 2017; Nair et al., 2018). For instance, it has been used to pre-train Q-functions by minimizing the temporal difference (TD) error between the demonstrated expert policy and the optimal policy (Hester et al., 2018), or to guide exploration through reward and policy shaping (Subramanian et al., 2016; Brys et al., 2015). Current behavior guidance methods strive to explore strategies beyond the original demonstrations (Gao et al., 2018; Brown et al., 2020), but generalization remains a fundamental challenge.

A primary issue of BC is the lack of generalization to unseen scenarios due to distributional mismatch (Ross et al., 2011). Given the high cost of collecting extensive demonstration data, efforts are made to learn generalizable policies with a limited number of demonstrations (Kim et al., 2013; Mandlekar et al., 2020; Prados et al., 2024). Several methods achieve better generalization by exploring perturbations in the policy parameter space (Such et al., 2017), known as evolutionary algorithms (Salimans et al., 2017). Further approaches add noise to the parameters and optimize through policy gradient descent to facilitate exploration (Plappert et al., 2017; Cao et al., 2020). Other works focus on generalizing to long-horizon, multi-stage tasks by leveraging the compositionality of a small set of given demonstrations (Mandlekar et al., 2020). They learn goal-directed policies from stochastic rollouts during the behavior cloning phase to achieve controllable behaviors. However, existing works still struggle with contextual variations and perform poorly in zero-shot settings.

**Hypernetworks** (Ha et al., 2017), generally speaking, use a neural network to generate the weights of a target network. During training, only the learnable hypernetwork weights are optimized. One of the noticeable merits of hypernetworks is the parameter efficiency. It achieves this by generating weights of multiple target networks on related tasks, performing well in soft-weight sharing (Chauhan et al., 2023b; Von Oswald et al., 2019; Navon et al., 2021; Zhao et al., 2020) and weight compression (Ha et al., 2017). Hypernetworks hold a good connection with meta-learning, primarily due to their intrinsic capability to facilitate the learning-to-learn process. Research such as (Sendera et al., 2023; Przewięźlikowski et al., 2022) employs hypernetworks to encapsulate the inner-update loop within the MAML framework (Mishra et al., 2017). These approaches help position hypernetworks as a meta-optimizer, a concept that has also inspired the methodology adopted in our study.

In the field of multi-task RL and meta-RL, which focus on policy generation and generalization, hypernetworks have achieved successful applications in areas such as morphology control (Xiong et al., 2023; 2024), episodic memory (BG et al., 2024), context-aware response (Beukman et al., 2024), and dynamics model generation (Xian et al., 2021). For instance, using an MLP-based hypernetwork to learn a universal policy for different robotic morphologies can maintain computational efficiency while achieving inference performance comparable to that of Transformers (Xiong et al., 2024). Other approaches demonstrate that when incorporating contextual information into policy adaptation, generating the weights of decision adapters with a hypernetwork can effectively circumvent the computational constraints caused by the expansion of learnable parameters (Beukman et al., 2024). As far as we observe, these methods (Xian et al., 2021; Beck et al., 2023) focus on the combination of the hypernetworks and zero-shot contextual RL. However, our method focuses on the hypernetwork structure and explicit modeling of the optimization procedure of the target network for achieving generalizable policy generation.

**Learned Optimizers** In this work, we design the hypernetwork to work like an optimizer. This is in concept similar to learned optimizers (LO) (Harrison et al., 2022; Bengio et al., 2013; Runarsson & Jonsson, 2000; Andrychowicz et al., 2016; Wichrowska et al., 2017). LO are models designed to enhance the efficiency and effectiveness of optimization processes. Unlike traditional hand-crafted algorithms such as SGD or Adam, LO leverages meta-learning to learn optimization strategies directly. Their primary goal is to generalize across tasks and outperform standard optimizers by adapting to specific problem structures. A key distinction is that LO is meta-learned and requires tuning on novel tasks. In contrast, our proposed structure predicts the update scheme directly from task specifications, eliminating the need for any data at test time.

Figure 2: (a) Given expert demonstrations $\mathcal{D}(\mathcal{M})$, one can always achieve the optimal policy $\theta^*$ via BC with stochastic gradient descent (SGD), which also defines the generalization upper bound in the context of BC for task $\mathcal{M}$. (b) Due to the lack of demonstrations, conventional hypernetworks generate parameters $\hat{\theta}$ for the task $\mathcal{M}$, by supervised learning the mapping from task to parameters, subject to overfitting given infinite solution space. (c) Even without actual demonstrations, we propose that the task $\mathcal{M}$ is informative of the data, and can thus be used to guide the iterative update for predicting the optimal parameters $\hat{\theta}^*$ (as in (a), Sec. 4.1), which in turn constrains the solution space and promotes prediction as optimization (from the task instead of data, Sec. 4.2). (d) With the proposed architectural inductive bias, the hypernetwork in (c) is indeed performing iterative optimization (action prediction loss is decreasing) as the iteration (block) number increases.[1]

## 3  POLICY GENERATION WITH HYPERNETWORKS

Given a Markov Decision Process (MDP) $\mathcal{M} = (\mathcal{S}, \mathcal{A}, \mathcal{T}, R, \gamma)$ where $\mathcal{S}$ is the state space, $\mathcal{A}$ is the action space, $\mathcal{T} : \mathcal{S} \times \mathcal{A} \to p(\mathcal{S})$ is the transition function, $R : \mathcal{S} \times \mathcal{A} \to \mathbb{R}$ is the reward function, and $\gamma \in [0, 1]$ is the discount factor, an optimal control policy $\pi : \mathcal{S} \to p(\mathcal{A})$ is supposed to maximize the expected (discounted) return in the context of reinforcement learning (RL). Typically, the policy network $\pi$ is a multi-layer perceptron (MLP) (Rakelly et al., 2019; Sarafian et al., 2021; Rezaei-Shoshtari et al., 2023; Beck et al., 2023). However, training a policy network with RL is challenging (Ibarz et al., 2021), e.g., due to large exploration space and optimization difficulties.

To resolve the training difficulties in RL, behavior cloning (BC) trains a policy network in a supervised manner with a dataset of expert demonstrations $\mathcal{D} = \{(s_t^i, a_t^i)\}_{i=1}^{N_d}$ with $N_d$ being the number of demonstrations and $s_t^i, a_t^i$ being the state and action at time $t$ respectively. The optimal policy can then be obtained by minimizing the action prediction loss $\mathcal{L}(\theta, \mathcal{D}) = \sum_{i,t} \ell(\pi(s_t^i; \theta), a_t^i)$, with $\ell$ as a discrepancy metric and $\theta \in \Theta$ as the parameters of policy network $\pi$. Despite the stability and efficiency of the training procedure, BC relies heavily on expert data and may not generalize to unseen task specifications. To generalize, one has to collect demonstrations covering all different situations, which is time-consuming and costly, especially for real-world applications.

**Problem formulation.**  We aim to maximize the utilization of the limited demonstrations for BC by developing a mechanism that can automatically synthesize the "optimal" policy network (parameters) from a novel specification of a given task. *Specifically*, for source task set $S$ and target task set $T$, we denote the source tasks characterized by the corresponding MDP as $\mathcal{M}_S = \{\mathcal{M}_j | \mathcal{M}_j = (\mathcal{S}_j, \mathcal{A}_j, \mathcal{T}_j, R_j, \gamma), j \in S\}$, and similarly the target tasks are $\mathcal{M}_T = \{\mathcal{M}_j | j \in T\}$, with $\mathcal{M}_S \cap \mathcal{M}_T = \varnothing$, e.g., differing in environmental settings or dynamics. Moreover, for source task $\mathcal{M}_j \in \mathcal{M}_S$, there exists a collection of demonstrations $\mathcal{D}_j$, whereas for the target tasks in $\mathcal{M}_T$, no demonstrations are available. Our goal is to train a network with $\Omega$ as parameters, $\mathcal{H}^\Omega : \mathcal{M} \to \Theta$, to produce the weights $\theta$ of the policy network $\pi$ after consuming the task specification $\mathcal{M}$, such that $\pi(\cdot; \mathcal{H}(\mathcal{M}))$ minimizes the BC loss derived with the demonstrations $\mathcal{D}(\mathcal{M})$.

Formally, the neural network $\mathcal{H}$ functions as a hypernetwork (Ha et al., 2017), and can be trained by minimizing the following:

$$\mathcal{L}(\Omega, \{(\mathcal{M}_j, \mathcal{D}_j)\}_{j \in S}) = \sum_j \sum_{i,t} \ell(\pi(s_t^{i,j}; \mathcal{H}(\mathcal{M}_j)), a_t^{i,j}), \tag{1}$$

where $j$ is the task (specification) index and $i, t$ go through different demonstrations and time steps within a task. After training, the hypernetwork $\mathcal{H}$ should be able to synthesize the optimal parameters for each source task (specification) in $\mathcal{M}_S$, i.e., $\mathcal{L}(\Omega, \{(\mathcal{M}_j, \mathcal{D}_j)\}_{j \in S})$ is minimal. Note that without demonstrations for the target tasks $\mathcal{M}_T$, the performance of the synthesized policy networks fully

---

[1]Note that the loss is computed with demonstrations, which is only used here for validation purposes.

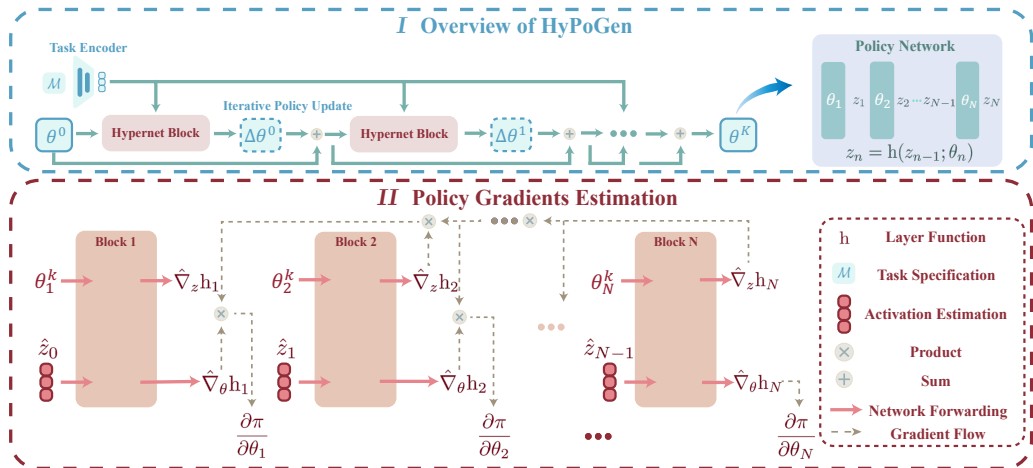

Figure 3: The overall pipeline of the proposed HyPoGen. Top: HyPoGen uses an iterative update scheme (Eq. 6) to generate the policy network parameters $\theta$ from the task specification $\mathcal{M}$. Bottom: inside each hypernet block, we calculate the neural "gradients" for each block of the policy network in a way resembling the backpropagation process (Eq. 7, 8).

depends on the generalization capability of $\mathcal{H}$ while only being trained on the source tasks $\mathcal{M}_S$ and the demonstrations $\{\mathcal{D}_j | j \in S\}$. Next, we elaborate on the generalization gap of $\mathcal{H}$ as a mapping from the task specification to the optimal parameters, and how we can improve the generalization by injecting an inductive bias into its structure that encourages optimization instead of memorization.

## 4    HYPERNETWORKS AS OPTIMIZATION WITHOUT DATA

**Motivation.**    As pointed out by (Heckel & Yilmaz, 2021; Nakkiran et al., 2021), an optimization process with a proper number of steps tends to yield parameters that generalize better than a model directly memorizing the training set. Currently, the architecture of hypernetworks is dominated by MLPs, especially those focusing on policy generation (Chauhan et al., 2023a). While an MLP is capable of mapping a task specification to the parameters of a policy network, it remains a very naive approach, offering limited generalization ability due to its lack of inductive bias. Building on this insight, we propose a novel hypernetwork to function as an iterative optimization process, rather than simply memorizing the best parameters for each training task. This operation is similar to the task-specific training with SGD and thus could potentially induce better generalization performance due to 1) this inductive bias constrains the search space for the hypernetwork, thus, less prone to overfitting to the training tasks; and 2) optimization is a low-degree-of-freedom operation and, if learned, will approach the upper bound set by task-specific training using demonstrations. Next, we will first introduce the designing principle in Sec. 4.1; and then we detail the proposed hypernetwork with the introduced bias in Sec. 4.2. An overview of the motivation is presented in Fig. 2, which also shows evidence that the proposed hypernetwork is performing optimization when checking the action prediction loss.

### 4.1    BIASING HYPERNETWORKS TOWARDS OPTIMIZATION

The overall pipeline of the proposed HyPoGen is illustrated in Fig. 3. In this section, we start by deriving the gradient formula for target networks on a given datapoint (Eq. 3, 4), and then we argue that the dependency on specific data can be resolved (Eq. 5). Finally, we present the model structure that incorporates the optimization inductive bias targeting MLP-based policy networks (Eq. 6, 7, 8).

Since most policy networks are MLPs due to their training efficiency and scalability, and without loss of generality, we can put a policy into the following format:

$$\pi(s; \theta) = h_N^{\theta_N} \circ h_{N-1}^{\theta_{N-1}} \circ h_{N-2}^{\theta_{N-2}} \circ \cdots \circ h_2^{\theta_2} \circ h_1^{\theta_1}(s), \tag{2}$$

where $\theta = \{\theta_n\}_{n=1}^N$ and $N$ is the number of layers (or blocks) in the policy network $\pi$. Please note that each layer $h$ does not need to be a fully connected layer, e.g., $h$ can be a block of layers or

a sub-network. The grouping of the parameters of a policy network into the format presented in Eq. 2 depends on its actual topology, and we mainly experiment with the MLP structure, given its popularity in the policy learning literature.

Furthermore, to fit a single data point $(s, a)$ under the $\ell_2$ distance, a gradient step updates the policy network parameters as:

$$\theta^{k+1} = \theta^k + \lambda \cdot \left( \pi(s; \theta^k) - a \right) \cdot \frac{\partial \pi(s; \theta^k)}{\partial \theta} \tag{3}$$

where, $\lambda \cdot (\pi(s; \theta^k) - a)$ denotes the effective stepsize and $\partial \pi(s; \theta^k)/\partial \theta$ can be computed using the chain rule:

$$\frac{\partial \pi(s; \theta^k)}{\partial \theta_n} = \left( \prod_{m=1}^{N-n} \frac{\partial \mathrm{h}_{N-m+1}(z_{N-m}; \theta_{N-m+1}^k)}{\partial z_{N-m}} \right) \frac{\partial \mathrm{h}_n(z_{n-1}; \theta_n^k)}{\partial \theta_n} , \tag{4}$$

with $z_n = \mathrm{h}(z_{n-1}; \theta_n)$ the activation of the $n$-th layer (block). Please note that even though Eq. 3 and Eq. 4 are instantiated with a single state-action pair, the counterparts computed with a dataset of demonstrations share a similar format. Therefore, we proceed to describe the proposed hypernetwork architecture without further laboring.

## 4.2 NEURAL ESTIMATION OF POLICY UPDATES WITHOUT DATA

Accurate computation of the updates in Eq. 3, 4 for deriving the optimal policy is difficult due to the lack of demonstration data for the target tasks, for example, both $\lambda \cdot (\pi(s; \theta^k) - a)$ and $\partial \pi(s; \theta^k)/\partial \theta$ in Eq. 3 are data-dependent. However, it is possible to approximate these updates (at least theoretically), if we treat the specification $\mathcal{M}$ of a target task as a sufficient representation of its demonstrations.

More specifically, the update of the policy parameters over all demonstrations is

$$\Delta \theta = \mathbb{E}_{(s,a) \sim p(\mathcal{D}(\mathcal{M}))} \left[ \lambda \cdot (\pi(s; \theta) - a) \cdot \frac{\partial \pi(s; \theta)}{\partial \theta} \right] = F(\theta, p(\mathcal{D}(\mathcal{M}))) , \tag{5}$$

Note that the expectation is taken over the distribution of training demonstrations $\mathcal{D}(\mathcal{M})$, which is defined by the task specification $\mathcal{M}$. The update of the policy parameters can be seen as a function $F$ that only depends on the policy network parameters $\theta$ and the distribution of the demonstrations $p(\mathcal{D}(\mathcal{M}))$.

With the above reasoning, it is sufficient to use an encoded representation of the task specification $\phi(\mathcal{M})$ to represent $p(\mathcal{D}(\mathcal{M}))$ and estimate the policy updates $\Delta \theta = F(\theta, \phi(\mathcal{M}))$. We use the same task encoder as in previous works (Rezaei-Shoshtari et al., 2023; Sarafian et al., 2021), though these works directly map the task specification to policy parameters. Taking this ground, we propose the following hypernetwork that leverages the structures presented in Eq. 3, 4 for promoting the optimization generalization, as well as tackling the lack of demonstration at test time.

**Iterative policy update with "neural gradients".** Inspired by Eq. 3, 5 and the above reasoning, we propose network modules $\lambda$ and $\psi$, such that the output of the proposed hypernetwork is as follows:

$$\mathcal{H}(\mathcal{M}) = \theta^K, \text{and } \theta^k = \theta^{k-1} + \lambda^k(\theta^{k-1}, \phi(\mathcal{M})) \cdot \psi^k(\theta^{k-1}, \phi(\mathcal{M})) , \tag{6}$$

where $\lambda^k$ and $\psi^k$ estimate the "stepsize" (i.e., $\lambda \cdot (\pi(s; \theta) - a)$) and "gradient" of the $k$-th policy update from the task embedding (specification) $\phi(\mathcal{M})$. The total number of updates is $K$, and $\theta^0$ is a set of learnable parameters. To improve the training efficiency, we also compress/recover the policy network parameters via an encoder and a decoder so that the hypernetwork $\mathcal{H}$ can perform the policy generation in a latent space. (details can be found in Sec. A.2).

We further incorporate the optimization inductive bias from Eq. 4 into the structure of $\psi$ to estimate the corresponding terms. Let $\nabla_z \mathrm{h}_n(z_{n-1}; \theta_n) = \partial \mathrm{h}_n(z_{n-1}; \theta_n)/\partial z_{n-1}$ and $\nabla_\theta \mathrm{h}_n(z_{n-1}; \theta_n) = \partial \mathrm{h}_n(z_{n-1}; \theta_n)/\partial \theta_n$ represent the gradients with respect to the activations and parameters, respectively. Then, we divide gradient estimation function $\psi$ into two sub-networks $\psi_n^z$ and $\psi_n^\theta$, implemented by MLPs, to estimate $\nabla_z \mathrm{h}_n, \nabla_\theta \mathrm{h}_n$ respectively. Moreover, we instantiate another series of MLPs $\phi_n^z, n = 1...N$ that compute the pseudo estimation of the activation $z_n$'s, i.e., $\hat{z}_n = \phi_n^z \circ \phi_{n-1}^z \circ \cdots \circ \phi_1^z(\phi(\mathcal{M}))$. One could also make the estimation of $\hat{z}_n$'s dependent on the current estimate of the latent parameters, but we omit the dependence for clarity and leave the necessity to experimental justifications.

With these, the neural estimate of $\nabla_z \mathrm{h}_n$ and $\nabla_\theta \mathrm{h}_n$ for each iteration $k$ are computed as:

$$\hat{\nabla}_z \mathrm{h}_n = \psi_n^z(\hat{z}_{n-1}, \theta_n) \text{ and } \hat{\nabla}_\theta \mathrm{h}_n = \psi_n^\theta(\hat{z}_{n-1}, \theta_n), \tag{7}$$

with the superscript $k$ omitted for clarity, and now we have the output of $\psi$ as:

$$\psi(\theta, \phi(\mathcal{M})) = \left\{ \left( \prod_{m=1}^{N-n} \hat{\nabla}_z \mathrm{h}_{N-m+1} \right) \hat{\nabla}_\theta \mathrm{h}_n \right\}_{n=1}^N, \tag{8}$$

which is the neural counterpart of the gradient computation illustrated in Eq. 4. Given these structural designs, we can denote the proposed policy-generation hypernetwork as $\mathcal{H} = \{\{\lambda^k\}_{k=1}^K, \phi, \{\phi_n^z, \psi_n^z, \psi_n^\theta\}_{n=1}^N, \theta^0\}$, which are the learnable parameters.

**Training loss.** Finally, we can train the proposed **hy**pernetwork $\mathcal{H}$ for **po**licy **gen**eration (HyPoGen) using the demonstrations of source tasks $\{\mathcal{M}_j, \mathcal{D}_j\}_{j \in S}$ by minimizing the BC loss presented in Eq. 1 in an end-to-end manner. Next, we study the effectiveness and generalization of the proposed hypernetwork in synthesizing policies for target tasks (from only the task specifications) without accessing any demonstrations.

## 5 EXPERIMENTS

The experiments are designed to evaluate the generalization capabilities of existing policy generation methods and to demonstrate the effectiveness of our proposed method, which biases the hypernetwork towards optimization. We leverage two sets of tasks for evaluation. The first set is derived from MuJoCo environments, which provide a robust platform for concept verification and are widely used by existing methods. The second set of tasks is sourced from the ManiSkill environment, which includes replicas of real robot arms and is more closely aligned with real-world scenarios. By conducting experiments in both MuJoCo and ManiSkill environments, we aim to comprehensively evaluate our methods, demonstrating their versatility in handling various types of task.

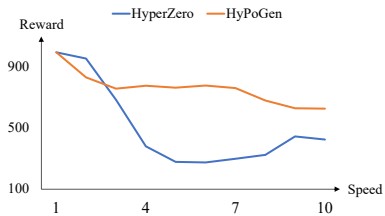

Figure 4: Our model shows better generalization than the baseline Hyper-Zero (details in Sec. 5.2).

Experiments are conducted individually on five selected representative tasks. Each task is varied by a group of specifications. It is important to emphasize that we focus on the generalization among different specifications within each task, rather than developing a generic policy generation for drastically different tasks. For each task, we split the specifications into a training set, denoted as $\mathcal{M}_S$, and a test set, denoted as $\mathcal{M}_T$, and $\mathcal{M}_S \cap \mathcal{M}_T = \varnothing$. The methods are trained on the expert trajectories from $\mathcal{M}_S$, and during test time, we roll out a policy for each specification in $\mathcal{M}_T$ and report the average of their performance (on the designated specification). Experimentally, instead of using the full MDP $\mathcal{M}$ as input, we omit the common component in $\mathcal{M}$ and use the task specification (length/target speed, etc.) as the input.

### 5.1 ENVIRONMENT SETUP

**MuJoCo.** Following HyperZero Rezaei-Shoshtari et al. (2023), we use three environments from DeepMind Control Suite Tunyasuvunakool et al. (2020) (a wrapper of the MuJoCo environment Todorov et al. (2012)) for evaluation: Cheetah, Walker, and Finger. For each task, we examine three types of specifications: (1) the desired speed of the object, (2) the torso length of the object, and (3) the combinations of speed and length.

**ManiSkill.** We utilize the ManiSkill2 environment Mu et al. (2021); Gu et al. (2023) to validate the real-world applicability. We select two tasks, LiftCube and Pick&PlaceCube, and conduct experiments using the Franka Panda robotic arm. We examine four types of specifications: (1) the size of the target cube, (2) the stiffness of the controller, (3) the damping of the controller, and (4) the arm length of the robot. The detailed settings of the task specifications can be found in Sec. B.1.

**Data collection and evaluation protocols.** For MuJoCo Environments, we pre-train a TD3 Fujimoto et al. (2018) agent on each specification separately for $1 \times 10^6$ steps as the expert. For each specification, 10 trajectories of length 1000 are collected as the demonstration. We randomly split the specifications into training and test sets using a **20%** to **80%** ratio. At test time, we evaluate the performance by calculating the **average reward** of roll-out policies. The training/test splitting process is repeated five times, and the average is calculated to reduce the effects of randomness.

For Maniskill Environments, we pre-train a PPO agent Schulman et al. (2017) for $4 \times 10^6$ steps and collect 1000 successful trajectories for each specification. We uniformly sample **30%** of the data for training, and use the remaining **70%** for testing. At test time, we calculate the **average success rate** and the **average episode length** for evaluation.

**Baselines.** For a fair comparison, all methods (including ours) use the same policy network architecture and specification encoder. The baselines are summarized as follows:

1. Specification conditioned policy (**Cond Policy**): This method aims to learn a general policy for all specifications. First, the network encodes the task specification using an encoder. It then takes the concatenation of the encoded specification and the state as input to predict the action.

2. Specification conditioned policy with **UVFA** Schaul et al. (2015): This method predicts actions and values under the UVFA framework. It first predicts the action in a similar way to specification-conditioned policy. Besides, it adds an extra branch to predict the action-value functions $Q(s, \hat{a})$ where $s$ is the current state and $\hat{a}$ is the predicted action, effectively utilizing the action-value in the expert rollouts as an extra supervision signal.

3. Specification conditioned **Meta Policy**: This method trains the policy network with the MAML Finn et al. (2017) framework. It aims to learn a set of weights that could adapt to different specifications. It predicts action similarly to specification-conditioned policy but trains differently through an inner-outer loop process, where the inner loop adapts the model to the support set, and the outer loop optimizes the model's initial parameters for rapid adaptation. Note that this method requires few-shot fine-tuning at test time.

4. **PEARL** Rakelly et al. (2019): Unlike other baselines, this method does not require access to contextual information. It instead keeps a replay buffer and uses it to generate an environment descriptor via a variational inference approach. This way, it can effectively leverage past experience and enable efficient off-policy learning. Note that this method also requires few-shot fine-tuning at test time.

5. **HyperZero** Rezaei-Shoshtari et al. (2023): It uses an MLP-based hypernetwork to predict the weight of policy networks. The hypernet takes the encoded specification as input and outputs the policy network parameters.

We conduct zero-shot policy generation evaluations for Cond Policy, UVFA, HyperZero, and our method. In contrast, the evaluations for Meta Policy and PEARL are carried out in a few-shot transfer setting. More specifically, these two methods are fine-tuned using expert trajectories from the test specifications $M_T$, thus breaking the no test-time demonstration assumption.

**Implementation details of HyPoGen.** We apply the same task encoder design as in Hyper-Zero (Rezaei-Shoshtari et al., 2023), which consists of 6 Res-Blocks. Following (Rezaei-Shoshtari et al., 2023), the policy networks are implemented as two-layer MLPs with hidden dimension 256 for MuJoCo and three-layer MLPs with hidden dimension 256 for ManiSkills. The input and output dimensions are specified by the task.

We use $K = 8$ hypernet blocks and apply Adam (Kingma & Ba, 2015) optimizer with learning rate $1 \times 10^{-4}$. We train our model on an NVIDIA 4090 GPU with a batch size of 512 for 2000 epochs (approximately 11 hours); we use the same hyperparameters in ManiSkill except for training 1000 epochs (approximately 10 hours). We detail our hyperparameters in Sec. B.2.

## 5.2 QUALITATIVE RESULTS

To intuitively illustrate the generalization ability of our method, we compare its performance with HyperZero (Rezaei-Shoshtari et al., 2023) on the Cheetah environment in MuJoCo. Both methods

Table 1: Comparison of the average reward achieved by different methods on the MuJoCo tasks. The **bold** and underline numbers indicate the best and second-best results.

| Reward ↑ | Cheetah | | | Finger | | | Walker | | |
|---|---|---|---|---|---|---|---|---|---|
| Method | speed | length | speed&length | speed | length | speed&length | speed | length | speed&length |
| Cond Policy | 433.24 | 574.70 | 356.68 | 379.66 | 409.30 | 289.58 | 151.95 | 336.71 | 186.15 |
| UVFA | 396.86 | 588.56 | 340.78 | 383.65 | 422.62 | 287.43 | 115.60 | 294.87 | 177.81 |
| Meta Policy | 337.35 | 579.41 | 199.63 | 125.89 | 217.58 | 114.78 | 44.54 | 46.22 | 44.44 |
| PEARL | 177.16 | 705.07 | 214.65 | 121.49 | 160.42 | 152.88 | 52.06 | 54.13 | 53.03 |
| HyperZero | 695.73 | 895.46 | 602.39 | 596.80 | 536.56 | 353.68 | 328.48 | 642.22 | 393.93 |
| HyPoGen(Ours) | **819.23** | **926.90** | **623.76** | **835.21** | **657.12** | **365.85** | **436.26** | **706.16** | **409.88** |
| Expert Rollout | 869.59 | 963.07 | 927.12 | 975.71 | 959.42 | 913.40 | 722.68 | 897.11 | 814.43 |

Table 2: Comparison of the success rate (%) and episode length (#steps) on the ManiSkill tasks. For each specification, we roll out 100 trajectories and compute the average success rate and episode length. Again, **bold** and underline numbers indicate the best and second-best results.

| % Rate ↑, #Steps ↓ | LiftCube | | | | Pick&PlaceCube | | | |
|---|---|---|---|---|---|---|---|---|
| Method | cube size | stiffness | damping | arm length | cube size | stiffness | damping | arm length |
| Cond Policy | 89.93, **32.09** | 0.00, 200.00 | 56.28, 94.27 | 69.36, 71.40 | 69.81, 75.06 | 2.20, 196.56 | 39.26, 131.41 | 43.77, 123.02 |
| UVFA | **90.27**, 32.40 | 0.00, 200.00 | 78.67, 53.53 | 72.18, 66.11 | 71.75, 71.91 | 0.00, 200.00 | 29.53, 146.68 | 36.85, 134.42 |
| Meta Policy | 84.67, 41.14 | 0.00, 200.00 | 0.06, 199.90 | 76.45, 56.95 | 12.62, 177.89 | 0.22, 199.77 | 0.21, 199.64 | 8.15, 185.68 |
| PEARL | 84.47, 41.60 | 59.87, 86.70 | 35.67, 132.23 | 78.73, 52.86 | 21.06, 163.28 | 24.44, 156.22 | 12.58, 177.54 | 16.08, 171.93 |
| HyperZero | 86.60, 38.24 | 0.00, 200.00 | 31.33, 140.90 | 63.64, 81.59 | 25.87, 155.27 | 0.00, 200.00 | 0.00, 200.00 | 13.92, 174.91 |
| HyPoGen(Ours) | 85.87, 39.37 | **97.20, 16.72** | **93.28, 24.78** | **85.73, 39.44** | **72.87, 68.97** | **78.33, 58.76** | **41.26, 125.99** | **52.54, 106.75** |
| Expert Rollout | 94.13, 22.86 | 96.73, 17.88 | 97.28, 16.30 | 93.64, 24.74 | 69.87, 73.26 | 75.92, 62.65 | 73.72, 66.19 | 54.77, 101.14 |

are trained on target speeds 1 and 10 and tested on speeds from 2 to 9, with reward curves shown in Fig. 4. These experiments show that even though the different tasks are varied by a single parameter (i.e., speed), transferring between them is challenging. While both methods perform similarly on training tasks, HyperZero's performance degrades significantly on unseen tasks with larger train-test gaps, whereas HyPoGen maintains stable performance, showing superior generalization.

## 5.3 COMPARISONS

**MuJoCo.** The test-time average rewards obtained from different methods are reported in Tab. 1. We can observe that the rewards obtained from conditioned policies (Cond Policy and UVFA) are considerably lower than those from the expert. This suggests that the effectiveness of using specifications as network input (condition) is limited. The two few-shot fine-tuning methods, Meta Policy and PEARL, also perform much worse than the expert rollout. HyperZero outperforms all other baselines, illustrating that hypernetwork is a more powerful policy generation paradigm than the conditioning methods. Our method yields the best results across all tasks and specifications, demonstrating the advantage of incorporating optimization inductive bias over direct policy generation. Please refer to Sec. C.1 for detailed results in each specification and Tab. 13 in the appendix for the standard deviation of the results.

**ManiSkill.** We report the average success rate and episode length in Tab. 2, and the standard deviation of episode length in Tab. 14 in the appendix. The max step of a trial is set to 200. All methods can achieve high success rates across different cube size specifications. We contend that this task is relatively simple, as it primarily involves controlling the clamp's opening width, which is nearly linear to the cube size. In more challenging scenarios where controller stiffness, damping, and arm length are varied, a linear relationship is no longer valid. As a result, the performances of different methods vary and degenerate. In the most challenging stiffness specifications, the conditional policy methods, the meta-learning baseline, and HyperZero fail to learn a reasonable policy. In contrast, the proposed method HyPoGen is able to generate policies that achieve high success rates and low episode length, demonstrating its superior generalization ability. Additionally, in-depth experiments in Sec. C.3 verified that the poor performance of the baselines is attributed not to a lack of capacity, but to their inductive bias, further highlighting HyPoGen's superior generalization.

## 5.4 ANALYSIS

In this section, we perform analysis and ablation studies to validate our model from different aspects. By default, we use the task Cheetah from MuJoCo, with different speed specifications.

**Does HyPoGen actually perform optimization?** Since we aim at biasing the hypernetwork towards optimization, one may be interested in knowing: (1) if HyPoGen simply remembers a fixed set of parameters for each specification, and (2) will the "neural gradients" work in a similar way to ordinary gradients? i.e., if they are really optimizing the BC loss.

Table 3: Statistics of the generated parameters by HyPoGen. We report the average magnitude of the parameters and their standard deviation.

|  | Magnitude | Std. |
|---|---|---|
| fc1.weight | 43.92 | 9.32 |
| fc1.bias | 6.17 | 2.20 |
| fc2.weight | 9.31 | 5.53 |
| fc2.bias | 0.44 | 0.31 |

For the first question, We fix the input specification $M$ and examine the response of HyPoGen with various input weights $\theta^0$. Specifically, we replace the values of $\theta^0$ with uniformly random values within the range of $[-0.1, 0.1]$, and then compare the outputs of $\theta^K$ for different initial values of $\theta^0$. The statistics of weights and biases in $\theta^K$ are reported in Tab. 3. It is clear that the differences are significant, indicating that HyPoGen is not simply memorizing a certain set of fixed parameters for a given task specification.

As for the second question, since HyPoGen requires significantly fewer update steps than conventional gradient descent (8 versus thousands), the "neural gradient"

Table 4: BC loss after each "gradient" update.

| #Updates | 1 | 2 | 3 | 4 | 5 | 6 | 7 | 8 |
|---|---|---|---|---|---|---|---|---|
| Loss@Cheetah | 71.49 | 61.13 | 46.30 | 40.64 | 21.95 | 11.21 | 3.19 | 1.61 |
| Loss@LiftCube | 9.041 | 5.202 | 2.972 | 2.68 | 1.418 | 0.948 | 0.644 | 0.123 |

encapsulates much more information than the local gradient. Therefore, it is not appropriate to directly compare the values of neural gradients with conventional ones. Instead, we report the BC loss after each optimization step (i.e. $\text{BCLoss}(\theta^k)$). The results in Tab. 4 show that BC loss is constantly decreasing after each update, verifying the effectiveness of HyPoGen in predicting neural "gradients" that optimize the weights of the network. Thus, HyPoGen not only generates different updates for different input weights, but also "optimizes" in the right direction.

**Does the optimization-based hypernetwork improve training convergence?** We compare our method with HyperZero, examining how much reward (test-time) they can achieve when trained with the same epochs, and how many epochs each requires to achieve the same reward. The results are reported in Tab. 5. Our method achieves a higher reward with the same number of epochs. Besides,

Table 5: Convergence property at certain epochs or reward.

|  | 250 epochs | 600 reward |
|---|---|---|
| HyperZero | 571.3 | 394 |
| HyPoGen | 732.9 | 74 |

our method requires much fewer epochs of training to reach a given reward value. These results demonstrate the superior training efficiency of our method. Due to limited space, please refer to Sec. C for more experimental analysis.

## 6 DISCUSSION

We propose a novel hypernetwork, HyPoGen, which explicitly models the gradient-based optimization process in the architecture for policy synthesis, capitalizing on the generalization capability of test-time optimization. Additionally, we introduce an effective building block to mimic the interdependencies of the gradient flow while training the policies. This structural design further enhances the interpretability and generalizability of the proposed hypernetwork. Through comprehensive benchmarking, we demonstrate the superiority of our method in terms of generalization under limited learning budgets (both demonstration and computation) and the effectiveness of each proposed component. Our work not only contributes to the development of more efficient and generalizable policy learning algorithms but also provides insights for future research in the area of hypernetworks. However, generating policies without seeing demonstration data for completely different tasks remains challenging and is not fully resolved by our method. Future work may explore different optimization techniques and more sophisticated building blocks that can better generalize across a more diverse range of tasks.

## 7 REPRODUCIBILITY STATEMENT

We have made significant efforts to ensure the reproducibility of our work. The code, datasets, and instructions necessary to replicate our experiments are publicly available at https://github.com/ReNginx/HyPoGen. We hope these resources will assist future research in both reproducing and building upon our work.

## 8 ACKNOWLEDGMENT

This work is supported by the Early Career Scheme of the Research Grants Council (grant # 27207224), the HKU-100 Award, a donation from the Musketeers Foundation, the HKU Seed Fund for PI Research, and in part by the JC STEM Lab of Autonomous Intelligent Systems funded by The Hong Kong Jockey Club Charities Trust. Difan Zou and Youyi Zheng would like to thank the support by the NSFC Grant (No. 62172363, No. 62306252) and Guangdong NSF Grant (No. 2024A1515012444). Siyan Dong would like to thank the Institute of Data Science at HKU for a postdoctoral scholarship. The authors would also like to thank the HKU-Shanghai ICRC for providing valuable computing resources.

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

# A  MORE DETAILS OF THE METHOD

## A.1  WHY AN MLP CAN BE APPLIED AS A GRADIENT APPROXIMATOR FOR MLPS

We show that $\frac{\partial \mathcal{L}}{\partial W}, \frac{\partial \mathcal{L}}{\partial b}$, where $W, b$ are the weight and bias of an MLP layer, are very close to the form of MLP in this section.

The forward process of an MLP layer is:

$$z_n = \sigma(W_n z_{n-1} + b_n) \tag{9}$$

where $z_n$ is the hidden variable, and $\sigma$ is the activation function.

By taking the derivatives, we have

$$\frac{\partial z_n}{\partial W_n} = \Lambda_n(z_{n-1})(z_{n-1})^T \tag{10}$$

$$\frac{\partial z_n}{\partial b_n} = \Lambda_n(z_{n-1}) \tag{11}$$

where

$$\Lambda_n(z_{n-1}) = \mathrm{diag}(\sigma'(W_n z_{n-1} + b_n)) \tag{12}$$

To get from $\frac{\partial z_n}{\partial W_n}$ to $\frac{\partial \mathcal{L}}{\partial W_n}$, we apply the chain rules.

$$\frac{\partial \mathcal{L}}{\partial W_n} = \frac{\partial \mathcal{L}}{\partial z_N} \frac{\partial z_N}{\partial z_{N-1}} \cdots \frac{\partial z_{n+1}}{\partial z_n} \frac{\partial z_n}{\partial W_n} \tag{13}$$

$$= \frac{\partial \mathcal{L}}{\partial z_N} \prod_{i=n+1}^{N} \frac{\partial z_i}{\partial z_{i-1}} \frac{\partial z_n}{\partial W_n} \tag{14}$$

And $\frac{\partial z_n}{\partial z_{n-1}}$ can also be calculated from the forward process

$$\frac{\partial z_n}{\partial z_{n-1}} = \Lambda_n(z_{n1})W_n \tag{15}$$

Note that (Eq. 12) is just the output of another MLP with a different final activation. And other operations in (Eq. 10, 11, 14, 15) are fundamentally linear in nature. This linearity implies that the forms of these gradient expressions bear a strong resemblance to those typically encountered in MLP. As mentioned above, it is reasonable to use an MLP as a gradient approximator for MLPs.

## A.2  OPTIMIZATION IN THE LATENT PARAMETER SPACE

To ease training and memory consumption, the proposed hypernetwork predicts the policy updates in a latent space. Specifically, we employ a layer-level encoder $E_{\theta_n}$ to compress each layer of the target network $\theta$ into a latent space before feeding them into the hypernet blocks. During the optimization process, parameters are consistently updated within the latent space. After the optimization process, a layer-level decoder $D_{\theta_n}$ is utilized to reconstruct the latent representation back into the parameter space of $\theta$. This approach ensures that the size of our optimization network parameters does not expand significantly with an increase in the target network parameters, thereby enhancing the model's applicability. For ease of understanding, we have not illustrated these less critical details in Fig. 3.

# B  ADDITIONAL EXPERIMENTAL DETAILS

## B.1  DETAILED SPECIFICATIONS OF EACH TASK

### B.1.1  MUJOCO

**Desired speed.**  Tab. 6 shows the desired speed specifications we used for each task. The default value is utilized in the experiments that involve changing torso length. Conversely, during the torso length experiments, the default value of the desired speed is applied.

Table 6: Details of desired speed specifications.

| Environment | Range | No. of Samples | Default Value |
|---|---|---|---|
| Cheetah | [-10, 0), (0, 10], increment 0.5 | 40 | +10 |
| Walker | [-5, 0), (0, 5], increment 0.25 | 40 | +1 |
| Finger | [-15, 0), (0, 15], increment 1.0 | 30 | +15 |

Table 7: Details of torso length specifications.

| Environment | Range | No. of Samples | Default Value |
|---|---|---|---|
| Cheetah | [0.3, 0.7], increment 0.01 | 41 | 0.5 |
| Walker | [0.1, 0.5], increment 0.01 | 41 | 0.3 |
| Finger | [0.1, 0.4], increment 0.01 | 31 | 0.16 |

Table 8: Details of speed and length specifications.

| Environment | Range of Speed Parameter | Range of Length Parameter | No. of Samples |
|---|---|---|---|
| Cheetah | [+1, +10], increment 1.0 | [0.3, 0.7], increment 0.05 | $10 \times 9$ grid |
| Walker | [+1, +5], increment 0.5 | [0.1, 0.5], increment 0.05 | $9 \times 9$ grid |
| Finger | [+1, +15], increment 1.0 | [0.1, 0.4], increment 0.05 | $15 \times 7$ grid |

**Torso length.** In all environments, the torso length parameter refers to the geometric size of the torso of the controlled object. Different sizes change the physical properties such as mass, center of gravity, etc. A complete list of torso lengths we used is shown in Tab. 7.

**Desired speed and torso length.** Now, we list all the parameters in the experiments of jointly changing speed and length parameters in Tab. 8.

### B.1.2 MANISKILL.

The specifications for ManiSkill are shown in Tab. 9. Note that the term "arm length" represents the ratio of the arm length to its original length.

Table 9: Details of task specifications in ManiSkill Environment.

| Specification | Range of Parameters | No. of Samples | Default Value |
|---|---|---|---|
| Cube Size | [0.01, 0.03], increment 0.001 | 21 | 0.02 |
| Controller Stiffness | [500, 1500], increment 50 | 21 | 1000 |
| Controller Damping | [50, 150], increment 10 | 21 | 100 |
| Arm Length | [0.5, 2.0], increment 0.1 | 16 | 1.0 |

### B.2 ADDITIONAL IMPLEMENTATION DETAILS

We implemented our method in PyTorchPaszke et al. (2019) and the hyperparameters are reported in Tab. 10, Tab. 11, and Tab. 12.

Table 10: Hyperparameters of TD3.

| Hyperparameter | Setting |
|---|---|
| Learning rate | $1 \times 10^{-4}$ |
| Optimizer | Adam |
| Mini-batch size | 256 |
| Actor update frequency | 2 |
| Target networks update frequency | 2 |
| Target networks soft-update | 0.01 |
| Target policy smoothing stddev. clip | 0.3 |
| Hidden dim. | 256 |
| Replay buffer capacity | $10^6$ |
| Discount | 0.99 |
| Seed frames | 4000 |
| Exploration steps | 2000 |
| Exploration stddev. schedule | linear(1.0, 0.1, 1e6) |

Table 11: Hyperparameters of PPO.

| Hyperparameter | Setting |
|---|---|
| Learning rate | $3 \times 10^{-4}$ |
| Optimizer | Adam |
| Mini-batch size | 400 |
| Gamma | 0.85 |
| Target KL | 0.09 |
| Value function coefficient | 0.5 |
| Clip Range | 0.2 |
| Lambda Gae | 0.95 |
| Max Grad Norm | 0.5 |

Table 12: Hyperparameters of HyPoGen.

| Hyperparameter | Setting |
|---|---|
| Learning rate | $1 \times 10^{-4}$ |
| Optimizer | Adam |
| Mini-batch size | 512 |
| Hidden dim. of target policy networks | 256 |
| Task embedding dim. | 256 |
| Weight embedding dim. | 256 |
| No. Layers $K$ | 8 |
| No. MLP layers in Hypernet Blocks | 2 |
| MLP hidden dim. in Hypernet Blocks | 128 |

# C    MORE COMPARISON AND ANALYSIS

## C.1    DETAILED COMPARISON ON EACH SPECIFICATION

We showcase the rewards and their standard deviations of the rollout policies in MuJoCo in Fig. 5, Fig. 6, and Fig. 7.

We also show the success rate curves of different specifications in ManiSkill in Fig. 8, 9

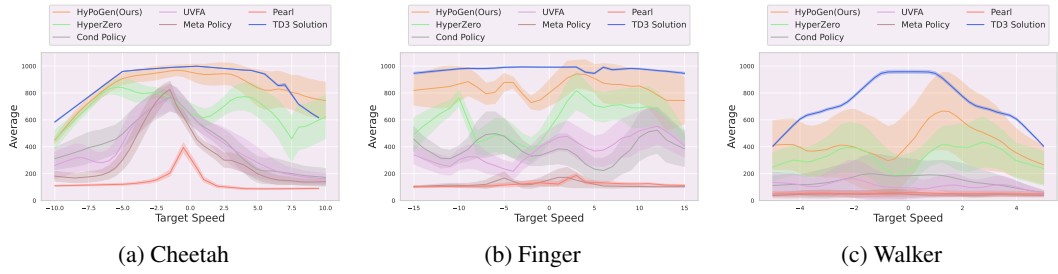

| (a) Cheetah | (b) Finger | (c) Walker |

Figure 5: Average reward and std curves on generalization to different **desired speed** specifications. The solid lines present the mean, and the shaded region presents the standard deviation. Our method achieves the best results in most cases.

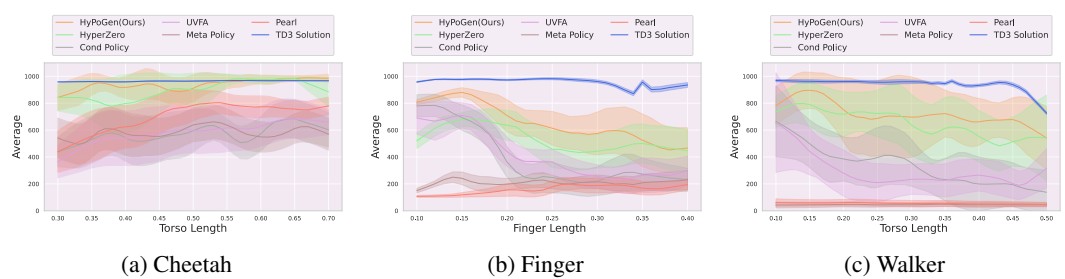

| (a) Cheetah | (b) Finger | (c) Walker |

Figure 6: Average reward and std curves on generalization to different **torso length** specifications. Our method achieves the best results in most cases.

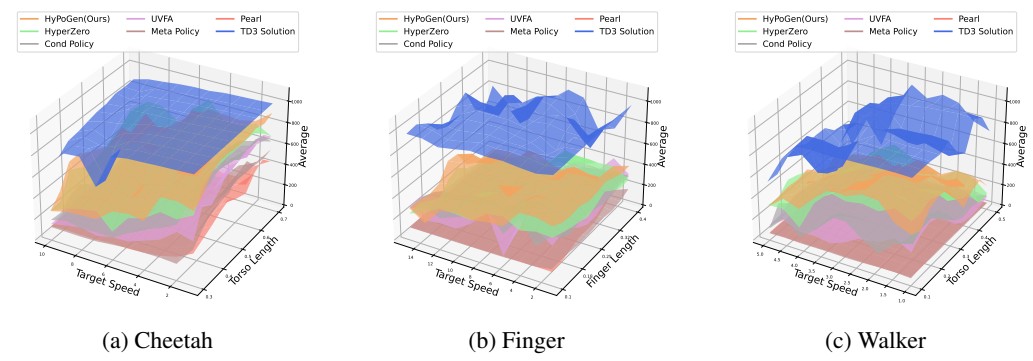

| (a) Cheetah | (b) Finger | (c) Walker |

Figure 7: 3D reward surfaces on generalization to different **speed and length** specifications. Our method achieves the best results in most cases.

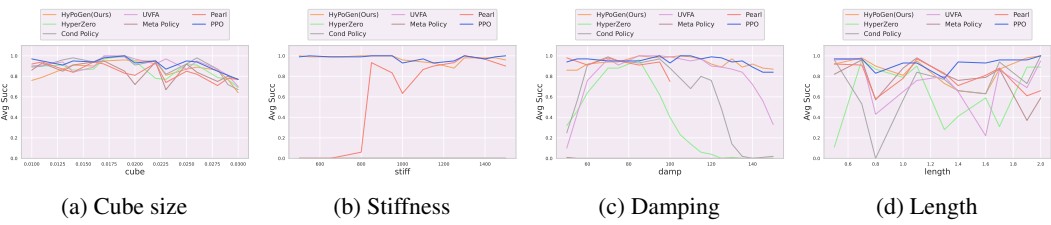

| (a) Cube size | (b) Stiffness | (c) Damping | (d) Length |

Figure 8: The success rate curve of different specifications on LiftCube task of Maniskill Environment.

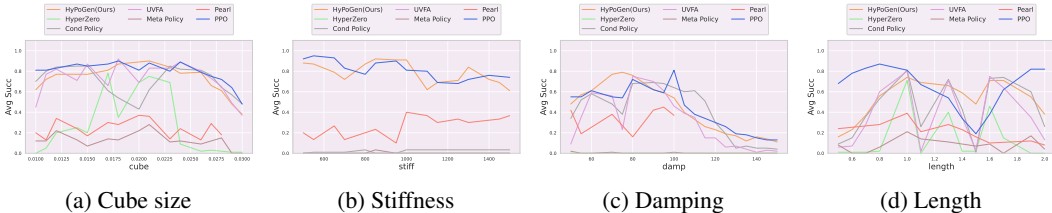

| (a) Cube size | (b) Stiffness | (c) Damping | (d) Length |

Figure 9: The success rate curve of different specifications on PickCube task of Maniskill Environment.

## C.2 STANDARD DEVIATION OF MAIN EXPERIMENTS

Below, we report the standard deviation of the rewards in MuJoCo environments in Tab. 13, and the standard deviation of episode length in ManiSkill in Tab. 14.

Table 13: Standard deviations on MuJoCo. The mean values are reported in Tab. 1 in the paper.

|  | Cheetah | | | Finger | | | Walker | | |
|---|---|---|---|---|---|---|---|---|---|
| **Method** | speed | lengths | speed&length | speed | lengths | speed&length | speed | lengths | speed&length |
| **Cond Policy** | 92.76 | 146.06 | 88.43 | 116.61 | 104.42 | 67.42 | 101.47 | 181.09 | 136.32 |
| **UVFA** | 86.59 | 165.48 | 84.68 | 100.03 | 111.45 | 63.04 | 79.65 | 145.64 | 125.10 |
| **Meta Policy** | 51.64 | 128.34 | 29.29 | 23.82 | 50.87 | 16.58 | 21.26 | 19.85 | 17.03 |
| **PEARL** | 19.39 | 118.11 | 30.77 | 11.33 | 38.79 | 33.16 | 20.57 | 21.67 | 21.83 |
| **HyperZero** | 84.08 | 103.30 | 137.36 | 107.90 | 115.00 | 80.67 | 140.51 | 195.85 | 191.87 |
| **HyPoGen** | 81.62 | 72.92 | 100.69 | 113.11 | 114.76 | 80.12 | 236.77 | 202.12 | 205.29 |
| **Expert Rollout** | 2.27 | 3.16 | 9.16 | 4.51 | 10.12 | 15.56 | 9.63 | 17.21 | 18.64 |

Table 14: Standard deviations of episode length on Maniskill. The mean values are reported in Tab. 2 in the paper.

|  | Lift Cube | | | | Pick&Place Cube | | | |
|---|---|---|---|---|---|---|---|---|
| **Method** | cube size | stiffness | damping | arm length | cube size | stiffness | damping | arm length |
| **Cond Policy** | 49.53 | 0.00 | 54.02 | 61.61 | 78.15 | 19.79 | 70.03 | 72.09 |
| **UVFA** | 49.57 | 0.00 | 52.15 | 68.64 | 74.70 | 0.00 | 64.10 | 65.76 |
| **Meta Policy** | 64.01 | 0.00 | 0.97 | 69.12 | 53.25 | 1.24 | 3.03 | 40.19 |
| **PEARL** | 65.42 | 38.89 | 18.40 | 70.52 | 64.23 | 71.22 | 30.36 | 52.70 |
| **HyperZero** | 58.62 | 0.00 | 47.44 | 68.76 | 51.38 | 0.00 | 0.00 | 35.00 |
| **HyPoGen** | 61.44 | 24.83 | 41.42 | 54.98 | 75.38 | 71.20 | 76.25 | 82.52 |
| **Expert Rollout** | 35.01 | 37.33 | 47.28 | 25.29 | 60.89 | 66.02 | 61.77 | 66.11 |

## C.3 ANALYSIS OF POOR PERFORMANCE OF BASELINES IN MANISKILL

To further analyze whether the poor performance of baselines is due to the lack of generalization ability or simply the model's capacity. First, we report the model size of each method in Tab. 15. Then, we report the original baselines' training performance in Tab. 16, and the training/test performance of the baselines with equal capacity to our models in Tab. 17. From these two tables, along with Tab.2, we can see that even without a capacity increase, some methods can already fit well on the training data. For example, HyperZero on the Pick&Place cube size task achieves a 78.20% success rate compared to 25.87% at test time, and its performance drops from 92% to 31% on the LiftCube damping task. Additionally, the performance of Cond Policy drops from 63% to 39%.

In theory, increasing the model capacity should make it easier to overfit. However, due to learning difficulty, the training performance actually decreased. For example, from Tab. 16 and Tab. 17, HyperZero dropped from 86% to 69% on the LiftCube arm-length task. When learning is not an issue, we observe that training performance improves, but generalization does not improve or even worsens. This is evidenced by Cond Policy's performance on the Pick&Place Cube arm-length task, where training performance increased from 47% to 61%, but test performance dropped from 43% to 26%.

These results and observations confirm that the proposed architecture design ensures the robustness of the learning process and eliminates the difficulty of training when capacity is increased. Moreover, it ensures that increased capacity does not incur overfitting and generalization issues, while this property is not observed with other models.

Table 15: Comparison of parameter sizes for different methods (Original vs Augmented), in Millions.

|  | Cond Policy | UVFA | Meta Policy | PEARL | HyperZero | HyPoGen(8 blocks) |
|---|---|---|---|---|---|---|
| **Original** | 0.7 | 0.7 | 0.7 | 0.7 | 25.1 | 70.0 |
| **Augmented** | 64.1 | 63.8 | 64.1 | 64.1 | 72.0 | 70.0 |

Table 16: Training-time success rate of baselines with original model size in Maniskill environments.

|  | Lift Cube | | | | Pick&Place Cube | | | |
|---|---|---|---|---|---|---|---|---|
| **Method** | cube size | stiffness | damping | arm length | cube size | stiffness | damping | arm length |
| **Cond Policy** | 94.67 | 0.00 | 98.33 | 89.80 | 86.60 | 2.56 | 63.00 | 46.67 |
| **UVFA** | 94.17 | 0.00 | 97.33 | 89.00 | 87.40 | 0.00 | 68.50 | 49.00 |
| **Meta Policy** | 83.83 | 0.00 | 0.00 | 77.20 | 17.00 | 0.56 | 0.00 | 11.33 |
| **PEARL** | 79.17 | 64.33 | 91.67 | 84.20 | 27.60 | 27.78 | 37.00 | 22.67 |
| **HyperZero** | 92.67 | 0.00 | 92.00 | 86.00 | 78.20 | 0.00 | 0.00 | 3.00 |
| **HyPoGen** | 89.33 | 96.50 | 96.33 | 85.60 | 82.40 | 81.33 | 68.00 | 68.67 |
| **Expert Rollout** | 96.00 | 95.33 | 90.75 | 98.50 | 87.00 | 82.33 | 58.25 | 84.00 |

Table 17: Training/test-time success rate of baselines with equal model size to ours in Maniskill environments.

|  | Lift Cube | | | | Pick&Place Cube | | | |
|---|---|---|---|---|---|---|---|---|
| **Method** | cube size | stiffness | damping | arm length | cube size | stiffness | damping | arm length |
| **Cond Policy** | 95.56 / 28.67 | 0.00 / 0.00 | 63.33 / 14.63 | 89.17 / 56.36 | 82.22 / 26.67 | 4.44 / 1.11 | 32.22 / 9.65 | 60.83 / 26.67 |
| **UVFA** | 95.56 / 28.89 | 0.00 / 0.00 | 64.44 / 16.48 | 83.33 / 58.18 | 81.11 / 23.96 | 4.44 / 1.78 | 15.56 / 7.02 | 38.33 / 18.21 |
| **Meta Policy** | 0.00 / 0.22 | 0.00 / 0.00 | 0.00 / 0.00 | 1.67 / 1.21 | 1.11 / 0.42 | 1.11 / 0.00 | 0.00 / 0.18 | 0.00 / 0.00 |
| **PEARL** | 0.00 / 0.00 | 0.00 / 0.22 | 0.00 / 0.74 | 5.00 / 1.21 | 0.00 / 0.62 | 2.22 / 0.00 | 0.00 / 0.88 | 0.83 / 0.00 |
| **HyperZero** | 94.44 / 28.00 | 0.00 / 0.00 | 63.33 / 22.22 | 69.17 / 32.73 | 77.78 / 25.62 | 0.00 / 0.00 | 0.00 / 0.00 | 21.67 / 8.46 |
| **HyPoGen** | 89.33 / 85.87 | 96.50 / 97.20 | 96.33 / 93.28 | 85.60 / 85.73 | 82.40 / 72.87 | 81.33 / 78.33 | 68.00 / 41.26 | 68.67 / 52.54 |
| **Expert Rollout** | 96.00 / 94.13 | 95.33 / 96.73 | 90.75 / 97.28 | 98.50 / 93.64 | 87.00 / 69.87 | 82.33 / 75.92 | 58.25 / 73.72 | 84.00 / 54.77 |

## C.4 ADDITIONAL ANALYSIS

**Scalibility: can HyPoGen generate deeper target networks?** We conducted experiments on deeper policy networks, and the results show that our method consistently outperforms the SOTA method HyperZero, as reported in Tab. 18.

In the comparisons section, we use two-layer and three-layer MLPs for MuJoCo and ManiSkill, respectively. Here, we gradually increase the number of layers in the policy networks. As the number of layers (i.e., the number of parameters to be generated by the hypernet) increases, the learning difficulty of the hypernet also increases. As observed from the table, our method consistently surpasses HyperZero in terms of obtained rewards at test time.

Table 18: Comparison of HyPoGen and Hyper-Zero with different layers in target networks.

| #Layers | 2 | 3 | 4 | 6 | 7 |
|---|---|---|---|---|---|
| HyperZero | 695.7 | 704.5 | 434.4 | 528.6 | 509.3 |
| HyPoGen | 819.2 | 821.9 | 819.1 | 665.6 | 622.5 |

**Generalization ability w.r.t. the amount of training data.** We compare our method with HyperZero to emphasize our generalization ability. In the experiment, we split 50% of the task specifications for testing and vary the fractions of the remaining specifications for training. The fractions range from 5% to 50%. The results are reported in Tab. 19. Our method achieves constantly better results than HyperZero. Note that as the fraction decreases, our method exhibits more significant improvements. It further demonstrates the strong generalization ability of our model.

Table 19: Zero-shot transfer with different amounts of training data.

|  | HyperZero | HyPoGen (ours) |
|---|---|---|
| 5% | 385.90 | 644.85 |
| 10% | 557.98 | 756.13 |
| 20% | 668.13 | 754.88 |
| 30% | 813.14 | 841.12 |
| 50% | 854.27 | 884.44 |

**Analysis on hyperparameters.** We show more results on the influence of hyperparameters in our model. We first validate the proper number of hypernet blocks (i.e., the optimization steps) in Tab. 20. We can observe that the performance reaches the peak when $K = 8$. In Tab. 21, we present the results of different MLP layers inside each hypernet block. The performance achieves the highest when the number of layers is set to 2. We also examine the feature dimensions of hidden layers in the hypernet block. The results are reported in Tab. 22. When setting the hidden dimension as 128, the network produces the best result.

Table 20: Performance with different optimization steps (No. of hypernet blocks).

| $K$ | Reward and Std. |
|---|---|
| 1 | $746.06 \pm 83.54$ |
| 3 | $775.55 \pm 68.26$ |
| 5 | $825.76 \pm 76.61$ |
| 8 | $\mathbf{856.88} \pm 61.73$ |
| 10 | $832.76 \pm 78.50$ |
| 20 | $817.9 \pm 60.92$ |

Table 21: Performance with different number of layers in each hypernet block.

| No. MLP Layers | Reward and Std. |
|---|---|
| 2 | $\mathbf{856.88} \pm 61.73$ |
| 3 | $795 \pm 70.39$ |
| 5 | $834.99 \pm 78.62$ |
| 8 | $752.74 \pm 119.31$ |
| 10 | $772.82 \pm 117.64$ |

Table 22: Performance with different hidden dimensions in hypernet block.

| Hidden Dimension | Reward and Std. |
|---|---|
| 16 | $847.83 \pm 59.95$ |
| 32 | $817.12 \pm 67.35$ |
| 64 | $840.02 \pm 61.24$ |
| 128 | $\mathbf{856.88} \pm 61.73$ |
| 256 | $834.5 \pm 80.36$ |

## C.5 STANDARD DEVIATION OF THE SUCCESS RATE ON MANISKILL

The standard deviations of success rate in the Manskill environment are shown in Tab. 23. The proposed method has the same deviation level as other baseline methods but is higher in the mean value.

Table 23: Average and standard deviations of success rate on Maniskill.

| Method | Lift Cube | | | | Pick&Place Cube | | | |
|---|---|---|---|---|---|---|---|---|
| | cube size | stiffness | damping | arm length | cube size | stiffness | damping | arm length |
| **Cond Policy** | $89.93 \pm 8.00$ | $0.00 \pm 0.00$ | $56.28 \pm 10.00$ | $69.36 \pm 14.00$ | $69.81 \pm 18.00$ | $2.20 \pm 2.00$ | $39.26 \pm 17.00$ | $43.77 \pm 17.00$ |
| **UVFA** | $90.27 \pm 8.00$ | $0.00 \pm 0.00$ | $78.67 \pm 10.00$ | $72.18 \pm 15.00$ | $71.75 \pm 17.00$ | $0.00 \pm 0.00$ | $29.53 \pm 14.00$ | $36.85 \pm 14.006$ |
| **Meta Policy** | $84.67 \pm 12.00$ | $0.00 \pm 0.00$ | $0.06 \pm 0.00$ | $76.45 \pm 15.00$ | $12.62 \pm 10.00$ | $0.22 \pm 0.00$ | $0.21 \pm 0.00$ | $8.15 \pm 7.00$ |
| **PEARL** | $84.47 \pm 12.00$ | $59.87 \pm 5.00$ | $35.67 \pm 3.00$ | $78.73 \pm 15.00$ | $21.06 \pm 15.00$ | $24.44 \pm 16.00$ | $12.58 \pm 8.00$ | $16.08 \pm 12.00$ |
| **HyperZero** | $86.60 \pm 11.00$ | $0.00 \pm 0.00$ | $31.33 \pm 10.00$ | $63.64 \pm 14.00$ | $25.87 \pm 10.00$ | $0.00 \pm 0.00$ | $0.00 \pm 0.00$ | $13.92 \pm 6.00$ |
| **HyPoGen** | $85.87 \pm 11.00$ | $97.20 \pm 3.00$ | $93.28 \pm 6.00$ | $85.73 \pm 10.00$ | $72.87 \pm 17.00$ | $78.33 \pm 15.00$ | $41.26 \pm 18.00$ | $52.54 \pm 20.00$ |

## C.6 STANDARD DEVIATIONS ON SUCCESS EPISODES

To align with established work Rezaei-Shoshtari et al. (2023), the result is shown in Tab. 1, 2 count in the failure episodes, which increases the variance reported in Tab.13, 14.

In Tab. 24, 25, we report the average of only success episodes[2]. As shown in the table, the standard deviations are of a reasonable scale relative to the averages.

Table 24: Average and standard deviations of rewards on MuJoCo on success episodes.

| | Cheetah | | | Finger | | | Walker | | |
|---|---|---|---|---|---|---|---|---|---|
| Method | speed | lengths | speed&length | speed | lengths | speed&length | speed | lengths | speed&length |
|---|---|---|---|---|---|---|---|---|---|
| **Cond Policy** | 510.50 | 609.70 | 370.11 | 378.38 | 441.97 | 301.73 | 604.32 | 315.07 | 384.91 |
| | ± 83.8 | ± 130.1 | ± 78.2 | ± 118.2 | ± 91.2 | ± 61.6 | ± 50.9 | ± 108.0 | ± 80.0 |
| **UVFA** | 452.58 | 657.83 | 358.22 | 385.83 | 462.36 | 292.53 | 504.84 | 244.98 | 370.07 |
| | ± 89.7 | ± 161.0 | ± 82.0 | ± 100.7 | ± 93.5 | ± 54.5 | ± 52.7 | ± 97.3 | ± 73.3 |
| **Meta Policy** | 355.62 | 697.48 | 232.19 | 122.97 | 174.37 | 123.83 | 84.56 | 92.13 | 81.74 |
| | ±44.3 | ±93.1 | ±30.8 | ±21.9 | ± 31.9 | ±20.6 | ± 13.3 | ± 6.2 | ± 9.6 |
| **PEARL** | 296.29 | 816.72 | 217.93 | 121.06 | 132.60 | 135.19 | 79.39 | 79.08 | 74.83 |
| | ± 22.6 | ± 69.1 | ± 25.1 | ± 11.2 | ± 30.6 | ± 26.9 | ± 6.8 | ± 10.8 | ± 8.0 |
| **HyperZero** | 772.16 | 917.14 | 626.06 | 618.75 | 565.97 | 384.97 | 764.24 | 508.74 | 537.99 |
| | ± 72.9 | ± 84.2 | ± 129.2 | ± 109.9 | ± 105.7 | ± 76.9 | ± 67.5 | ± 80.5 | ± 112.6 |
| **HyPoGen** | **845.81** | **949.86** | **653.53** | **854.99** | **697.78** | **396.76** | **839.94** | **636.54** | **601.73** |
| | **± 67.7** | **± 49.7** | **± 91.4** | **± 111.8** | **± 103.2** | **± 79.2** | **± 92.9** | **± 66.1** | **± 92.0** |

Table 25: Average and standard deviations of episode length on Maniskill on success episodes.

| | Lift Cube | | | | Pick&Place Cube | | | |
|---|---|---|---|---|---|---|---|---|
| Method | cube size | stiffness | damping | arm&length | cube size | stiffness | damping | arm&length |
|---|---|---|---|---|---|---|---|---|
| **Cond Policy** | 13.46 ± 7.09 | 0.00 ± 0.00 | 0.00 ± 0.00 | 0.00 ± 0.00 | 21.03 ± 10.13 | 0.00 ± 0.00 | 32.57 ± 14.27 | 26.22 ± 18.96 |
| **UVFA** | 14.56 ± 9.89 | 0.00 ± 0.00 | 14.10 ± 2.23 | 15.99 ± 6.73 | 21.80 ± 9.80 | 0.00 ± 0.00 | 25.33 ± 12.54 | **23.28 ± 11.84** |
| **Meta Policy** | **12.45 ± 4.14** | 0.00 ± 0.00 | 0.00 ± 0.00 | 13.11 ± 2.58 | 0.00 ± 0.00 | 0.00 ± 0.00 | 0.00 ± 0.00 | 0.00 ± 0.00 |
| **PEARL** | 12.60 ± 5.91 | 0.00 ± 0.00 | 126.11 ± 0.79 | 13.28 ± 4.22 | 46.72 ± 13.28 | 33.36 ± 8.42 | 134.20 ± 4.71 | 65.95 ± 11.32 |
| **HyperZero** | 13.31 ± 7.99 | 0.00 ± 0.00 | 0.00 ± 0.00 | 14.49 ± 3.82 | 0.00 ± 0.00 | 0.00 ± 0.00 | 0.00 ± 0.00 | 0.00 ± 0.00 |
| **HyPoGen** | 13.05 ± 7.65 | **11.41 ± 1.56** | **12.18 ± 2.14** | **12.68 ± 3.67** | **20.43 ± 9.94** | **19.55 ± 9.35** | **23.98 ± 11.78** | 23.46 ± 14.02 |

## C.7 IN AND OUT-OF-TRAINING-DISTRIBUTION PERFORMANCE

To better evaluate the generalization ability of the proposed methods, we evaluate all methods within one standard deviation of training distribution (in-distribution) and outside of that (out-of-distribution). With the result shown in Tab. 26, 27, 28, 29. Our proposed method exhibits superior performance over baseline methods in both in-distribution and out-distribution settings. This further highlights the strong generalization ability of the proposed method.

Table 26: The in-distribution average and standard deviations of rewards on MuJoCo.

| | Cheetah | | | Finger | | | Walker | | |
|---|---|---|---|---|---|---|---|---|---|
| Method | speed | lengths | speed&length | speed | lengths | speed&length | speed | lengths | speed&length |
|---|---|---|---|---|---|---|---|---|---|
| **Cond Policy** | 521.81 | 627.72 | 330.69 | 399.36 | 433.41 | 318.38 | 324.24 | 654.04 | 400.58 |
| | ± 85.4 | ± 131.5 | ± 70.6 | ± 129.4 | ± 102.2 | ± 59.1 | ± 57.6 | ± 110.1 | ± 84.3 |
| **UVFA** | 459.27 | 647.27 | 332.11 | 433.02 | 466.93 | 318.38 | 244.27 | 519.03 | 378.95 |
| | ± 85.3 | ± 157.6 | ± 78.2 | ± 109.8 | ± 96.6 | ± 59.1 | ± 48.3 | ± 104.5 | ± 72.7 |
| **Meta Policy** | 308.15 | 702.66 | 188.45 | 126.04 | 165.49 | 122.81 | 88.71 | 86.05 | 82.95 |
| | ± 39.3 | ± 89.4 | ± 25.8 | ± 24.4 | ± 34.2 | ± 19.7 | ± 10.4 | ± 6.4 | ± 11.0 |
| **PEARL** | 251.10 | 819.34 | 210.16 | 118.19 | 137.03 | 130.73 | 78.37 | 79.31 | 77.15 |
| | ± 18.0 | ± 70.2 | ± 21.7 | ± 10.8 | ± 35.7 | ± 25.8 | ± 6.25 | ± 10.8 | ± 8.6 |
| **HyperZero** | 796.92 | 941.41 | 640.15 | 691.46 | 585.75 | 446.38 | 555.23 | **792.82** | 596.58 |
| | ± 67.7 | ± 72.1 | ± 136.9 | ± 121.6 | ± 115.5 | ± 92.2 | ± 74.8 | **± 79.8** | ± 108.7 |
| **HyPoGen** | **837.94** | **956.45** | **653.97** | **880.00** | **706.10** | **459.51** | **621.51** | 706.10 | **645.18** |
| | **± 63.5** | **± 44.8** | **± 90.8** | **± 103.1** | **± 106.3** | **± 93.5** | **± 95.2** | ± 106.3 | **± 97.5** |

---

[2]MuJoCo does not provide a success indicator, thus when the reward of each frame is less than 0.05 for 50 consecutive frames, it is considered a failure.

Table 27: The out-of-distribution average and standard deviations of rewards on MuJoCo.

| Method | Cheetah | | | Finger | | | Walker | | |
|---|---|---|---|---|---|---|---|---|---|
| | speed | lengths | speed&length | speed | lengths | speed&length | speed | lengths | speed&length |
| **Cond Policy** | 488.88 | 574.32 | 427.34 | 343.66 | 460.60 | 236.09 | 294.89 | 486.90 | 321.78 |
| | ± 80.8 | ± 127.2 | ± 89.3 | ± 99.8 | ± 67.3 | ± 44.6 | ± 36.2 | ± 103.1 | ± 62.6 |
| **UVFA** | 440.42 | 676.61 | 394.64 | 308.00 | 453.66 | 236.09 | 247.72 | 477.46 | 340.54 |
| | ± 97.8 | ± 167.2 | ± 87.2 | ± 85.7 | ± 87.6 | ± 44.6 | ± 69.2 | ± 83.5 | ± 75.6 |
| **Meta Policy** | 429.27 | 687.59 | 297.92 | 117.87 | 190.76 | 125.88 | 98.18 | 81.86 | 78.68 |
| | ± 52.0 | ± 100.2 | ± 38.3 | ± 17.8 | ± 27.7 | ± 22.3 | ± 18.3 | ± 5.8 | ± 5.9 |
| **PEARL** | 374.23 | 811.10 | 229.90 | 125.88 | 124.18 | 144.47 | 80.16 | 79.56 | 70.50 |
| | ± 30.5 | ± 66.7 | ± 30.3 | ± 11.9 | ± 21.1 | ± 29.1 | ± 7.55 | ± 10.8 | ± 7.0 |
| **HyperZero** | 721.07 | 866.78 | 605.16 | 486.52 | 530.86 | 253.91 | 380.68 | **702.61** | 400.35 |
| | ± 83.6 | ± 109.4 | ± 117.9 | ± 88.6 | ± 88.3 | ± 44.1 | ± 47.2 | ± 82.1 | ± 121.8 |
| **HyPoGen** | **857.70** | **936.57** | **652.87** | **812.36** | **682.02** | **260.78** | **663.83** | 682.02 | **487.32** |
| | **± 74.0** | **± 59.5** | **± 92.3** | **± 126.6** | **± 97.4** | **± 48.3** | **± 88.7** | ± 97.4 | **± 77.4** |

Table 28: The in-distribution success rate and espisode length on Maniskill.

| Method | Lift Cube | | | | Pick&Place Cube | | | |
|---|---|---|---|---|---|---|---|---|
| | cube size | stiffness | damping | arm&length | cube size | stiffness | damping | arm&length |
| **Cond Policy** | **93.42, 12.77** | 0.00, 0.00 | 95.00, 10.89 | 66.30, 13.45 | 72.20, 21.24 | 2.19, 41.00 | 60.75, 17.82 | 61.00, 23.82 |
| **UVFA** | 92.75, 13.83 | 0.00, 0.00 | 92.80, 10.39 | 69.90, 16.11 | 80.20, 21.87 | 0.00, 0.00 | 65.00, 16.50 | 52.00, 20.21 |
| **Meta Policy** | 85.58, 11.65 | 0.00, 0.00 | 0.00, 0.00 | 78.20, 13.13 | 15.10, 22.86 | 0.28, 8.00 | 0.25, 5.75 | 11.50, 20.12 |
| **PEARL** | 85.00, 12.01 | 59.56, 11.34 | 93.80, 10.20 | 80.00, 13.37 | 25.70, 23.31 | 25.15, 21.50 | 32.00, 21.27 | 25.60, 22.87 |
| **HyperZero** | 88.58, 12.83 | 0.00, 0.00 | 79.60, 10.90 | 61.10, 14.55 | 40.20, 32.40 | 0.00, 0.00 | 0.00, 0.00 | 21.33, 14.13 |
| **HyPoGen** | 89.25, 12.64 | **97.00, 11.46** | **95.60, 10.53** | **84.30, 12.57** | **81.90, 19.75** | **79.75, 19.81** | **69.00, 17.67** | **65.17, 21.15** |

Table 29: The out-of-distribution success rate and episode length on Maniskill.

| Method | Lift Cube | | | | Pick&Place Cube | | | |
|---|---|---|---|---|---|---|---|---|
| | cube size | stiffness | damping | arm&length | cube size | stiffness | damping | arm&length |
| **Cond Policy** | 76.00, 16.23 | 0.00, 0.00 | 41.38, 14.92 | 100.00, 14.06 | **65.83, 20.68** | 2.22, 24.33 | 33.53, 36.51 | 29.00, 28.28 |
| **UVFA** | 80.33, 17.49 | 0.00, 0.00 | 73.23, 15.52 | 95.00, 14.74 | 57.67, 21.67 | 0.00, 0.00 | 20.07, 27.68 | 23.86, 25.92 |
| **Meta Policy** | 81.00, 15.63 | 0.00, 0.00 | 0.08, 1.85 | 59.00, 12.92 | 8.50, 16.80 | 0.00, 0.00 | 0.20, 4.40 | 5.29, 19.12 |
| **PEARL** | **82.33, 14.95** | 61.11, 6.73 | 86.50, 9.54 | 66.00, 12.33 | 20.00, 28.62 | 30.00, 21.28 | **35.75, 21.49** | 16.20, 28.61 |
| **HyperZero** | 78.67, 15.25 | 0.00, 0.00 | 12.77, 10.79 | 89.00, 13.82 | 2.00, 9.40 | 0.00, 0.00 | 0.00, 0.00 | 7.57, 15.36 |
| **HyPoGen** | 72.33, 14.71 | **98.00, 11.21** | **92.38, 12.81** | **100.00, 13.75** | 57.83, 21.57 | **72.67, 18.52** | 33.87, 25.66 | **41.71, 25.45** |

# D VISUALIZATION

Due to the page limit, we only present quantitative evaluations in the experiment section. Here, we visualize the results in the simulation environments and qualitatively compare our method with the baselines. The visual results of MuJoCo tasks are shown in Fig. 10, 11, 12, 13, 14, and 15. As seen in Fig. 10, our method can obtain more stable results with different speeds. When the target speed is -5.0 or -8.0 (the first and second rows), our method has a faster actual speed (better performance). Meanwhile, HyperZero is unable to maintain balance when the speed is 5.0 (the third row). In Fig. 11, HyperZero struggles to maintain balance, while our method generates reasonable results. In the Finger Spin environment in Fig 12, HyperZero stagnates for a long time when the speed equals -5.0 and 5.0. On the contrary, our method works well for all settings, demonstrating its generalization ability and stability. The same as the results of different speeds, in different torso lengths, the results (in Fig. 13, 14, and 15) demonstrate generalization and stability of our method. Our method encounters fewer failures. and also guides the agents to act at a faster speed. The visual results on ManiSkill tasks are shown in Fig 16, 17, 18, 19, 20, 21, 22, and 23. Please turn to the next page to see the visual results.

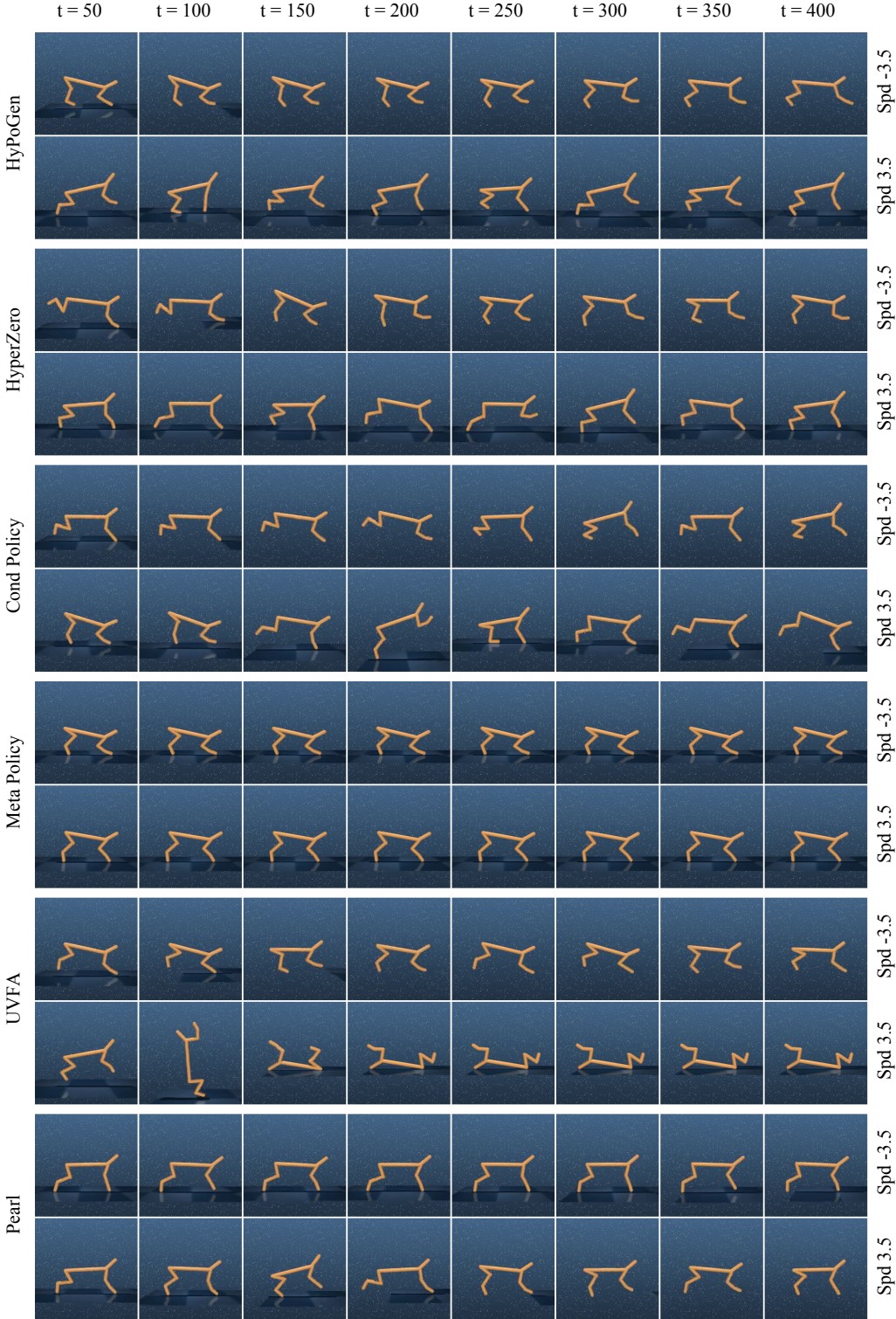

Figure 10: Comparison in Cheetah environment with different speeds.

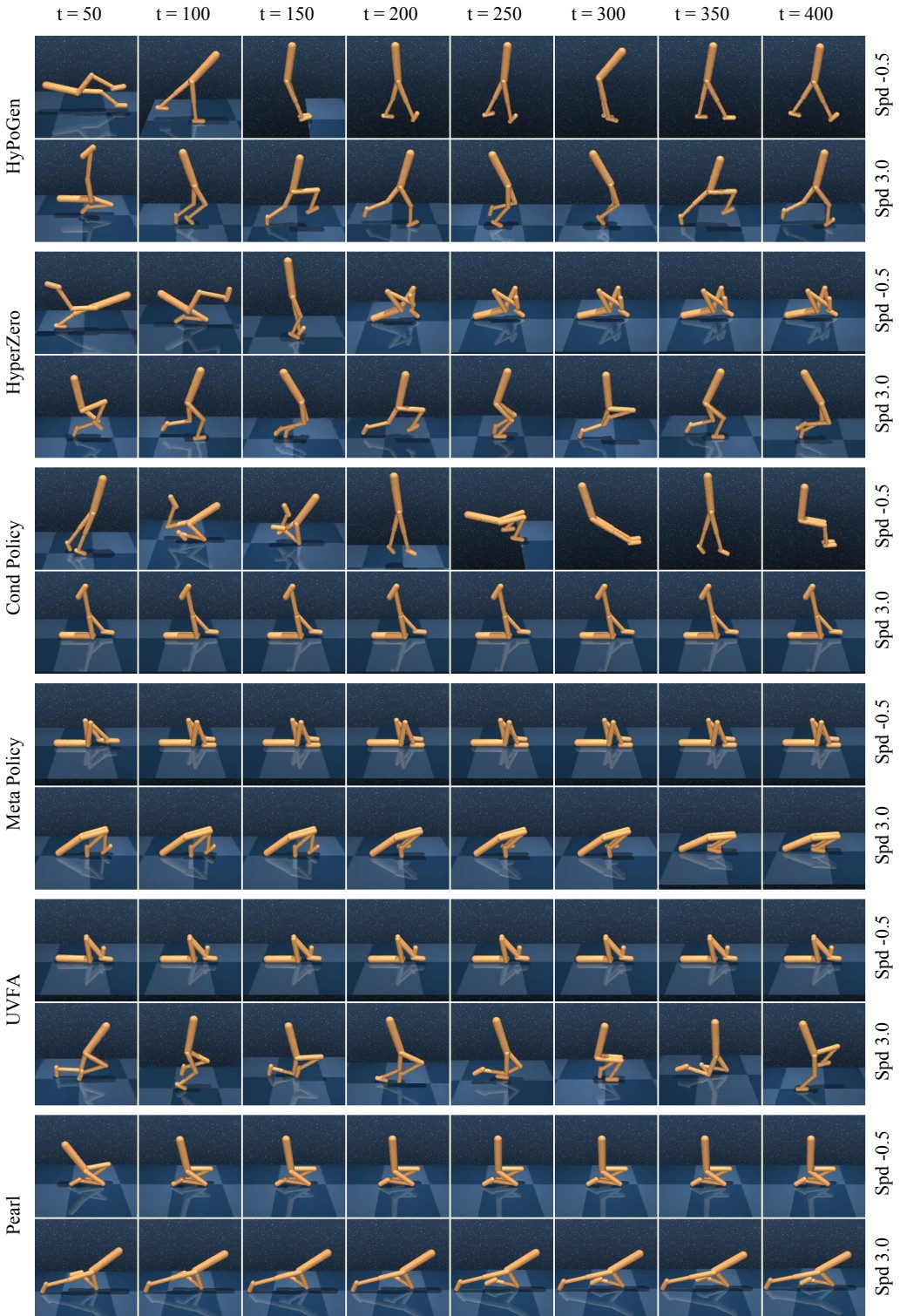

Figure 11: Comparison in Walker environment with different speeds.

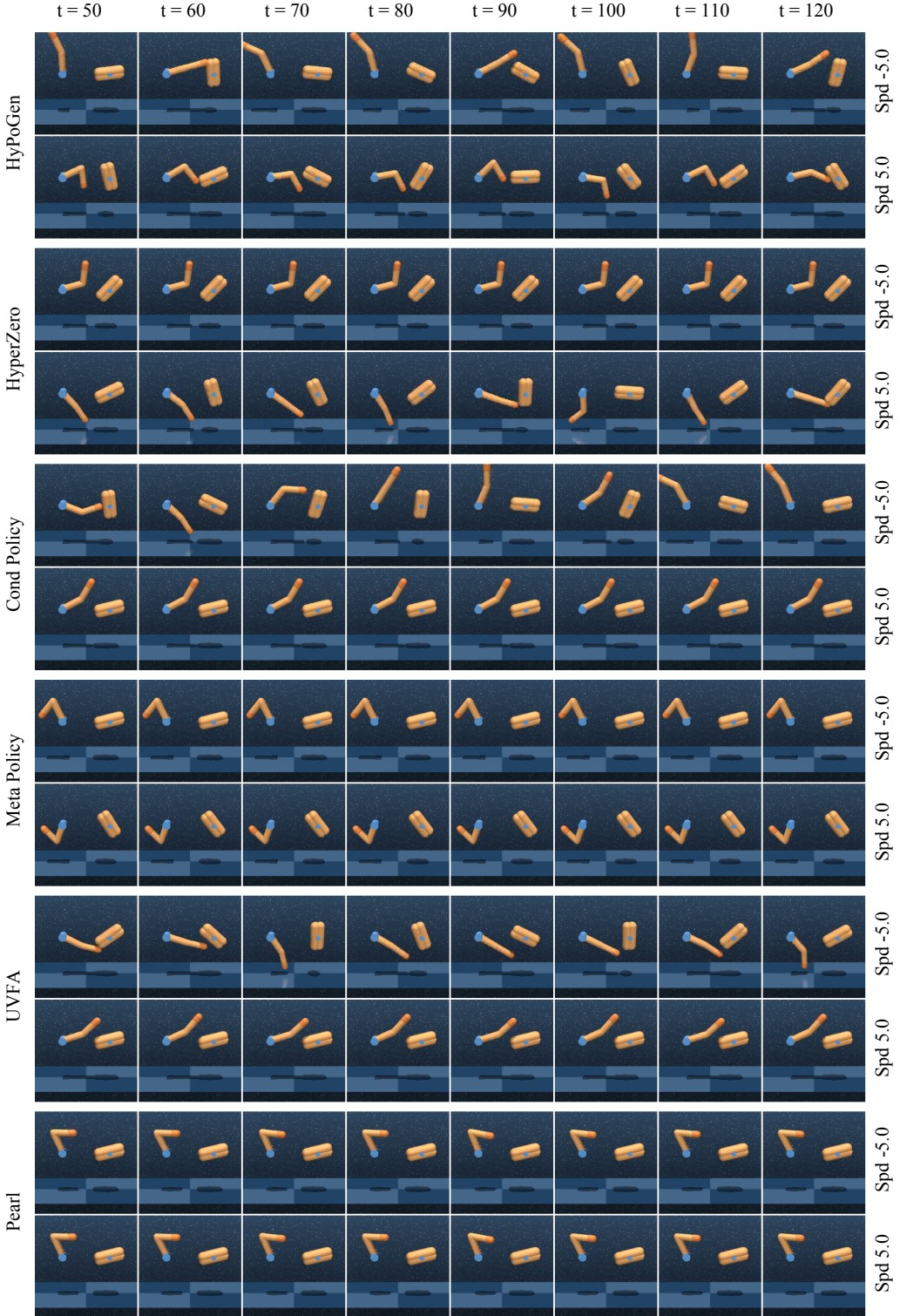

Figure 12: Comparison in Finger environment with different speeds.

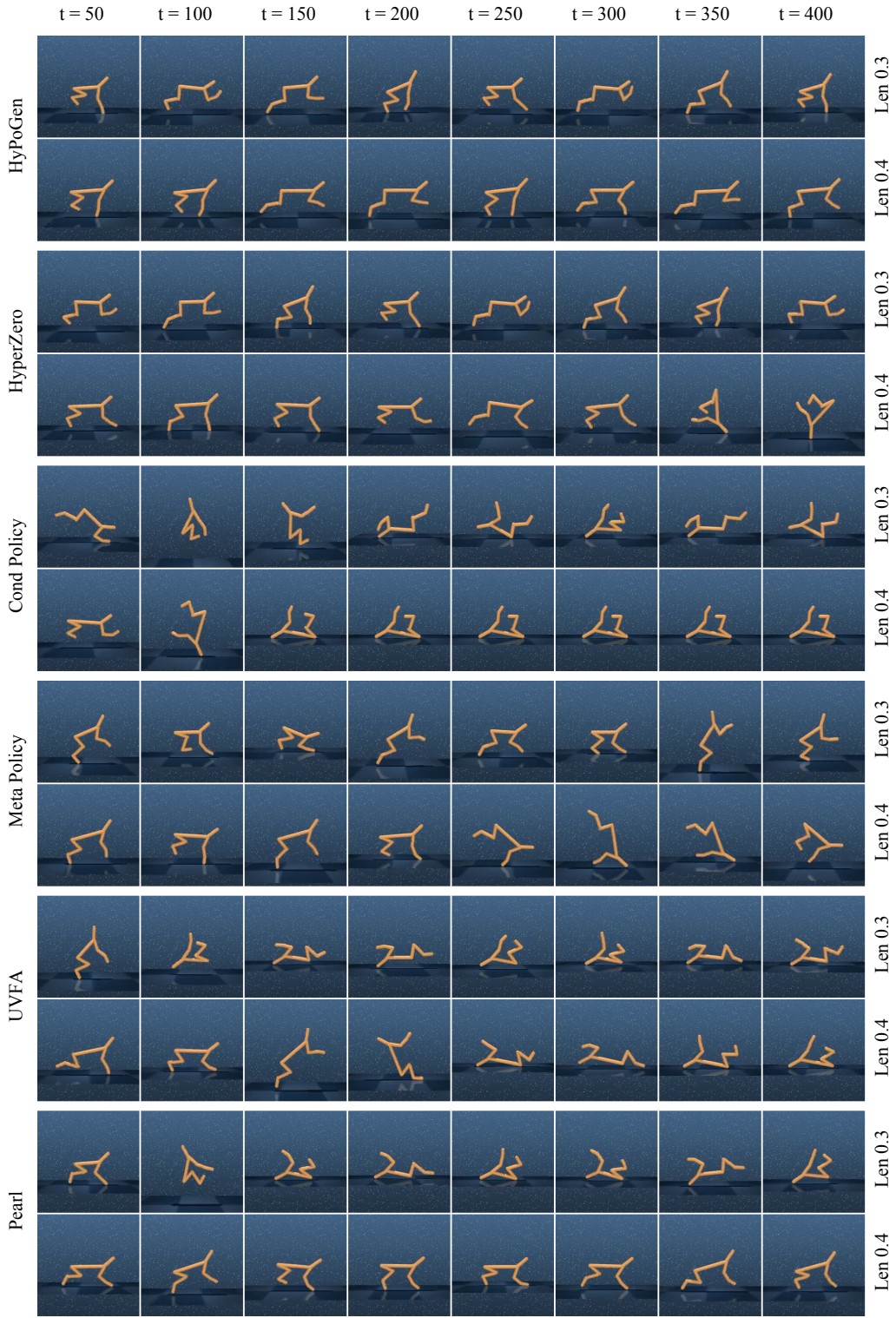

Figure 13: Comparison in Cheetah environment with different torso lengths.

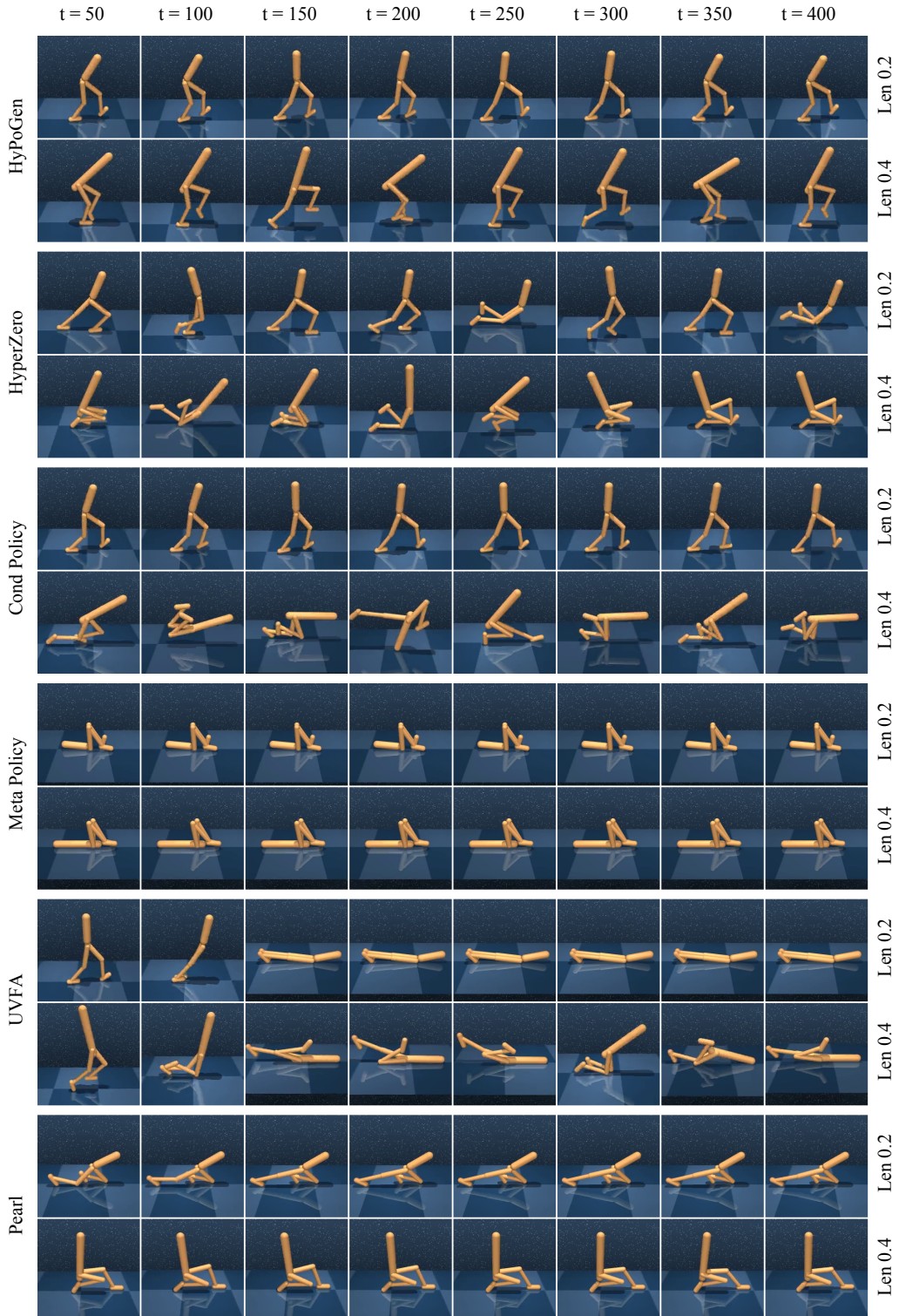

Figure 14: Comparison in Walker environment with different torso lengths.

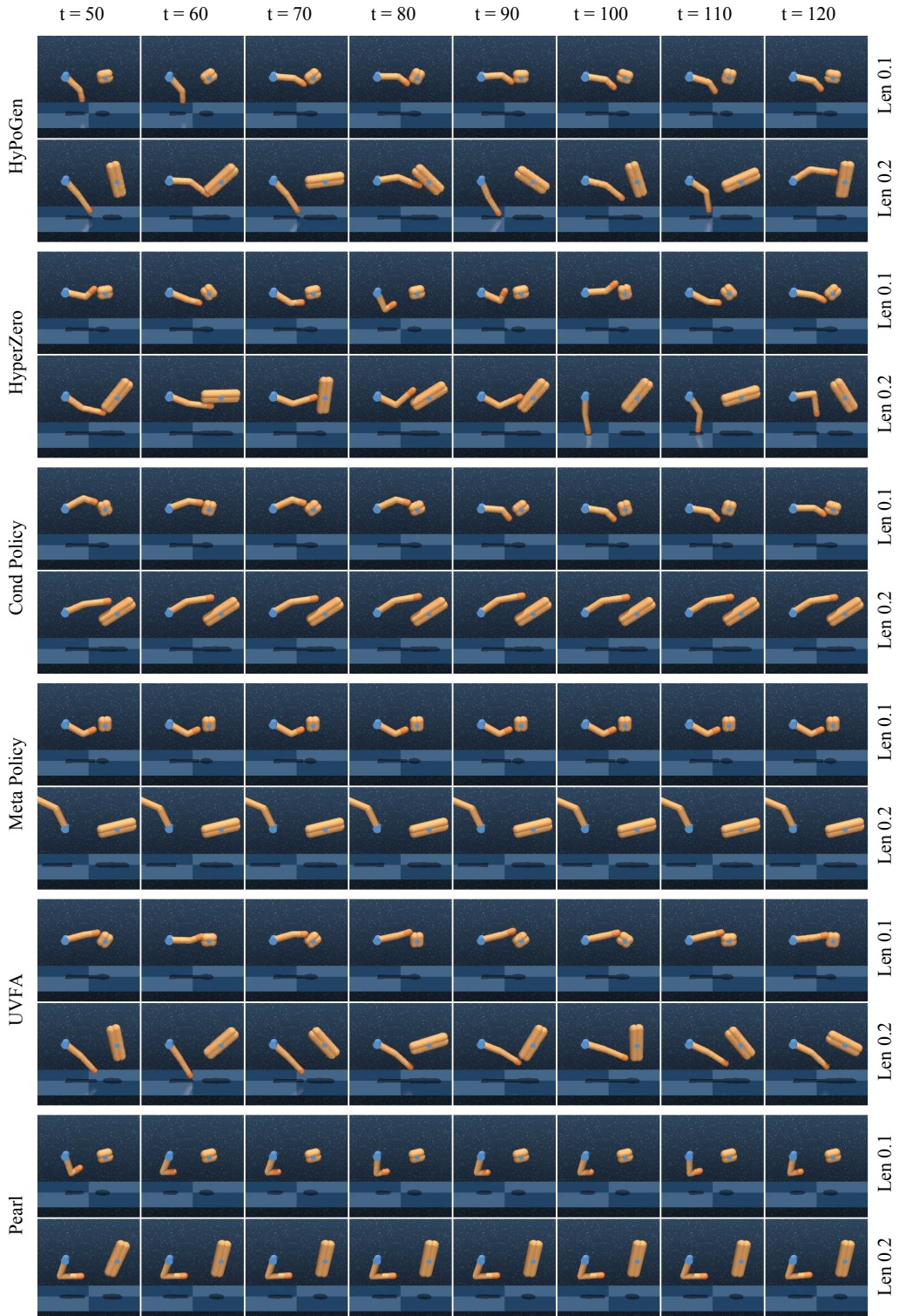

Figure 15: Comparison in Finger environment with different torso lengths.

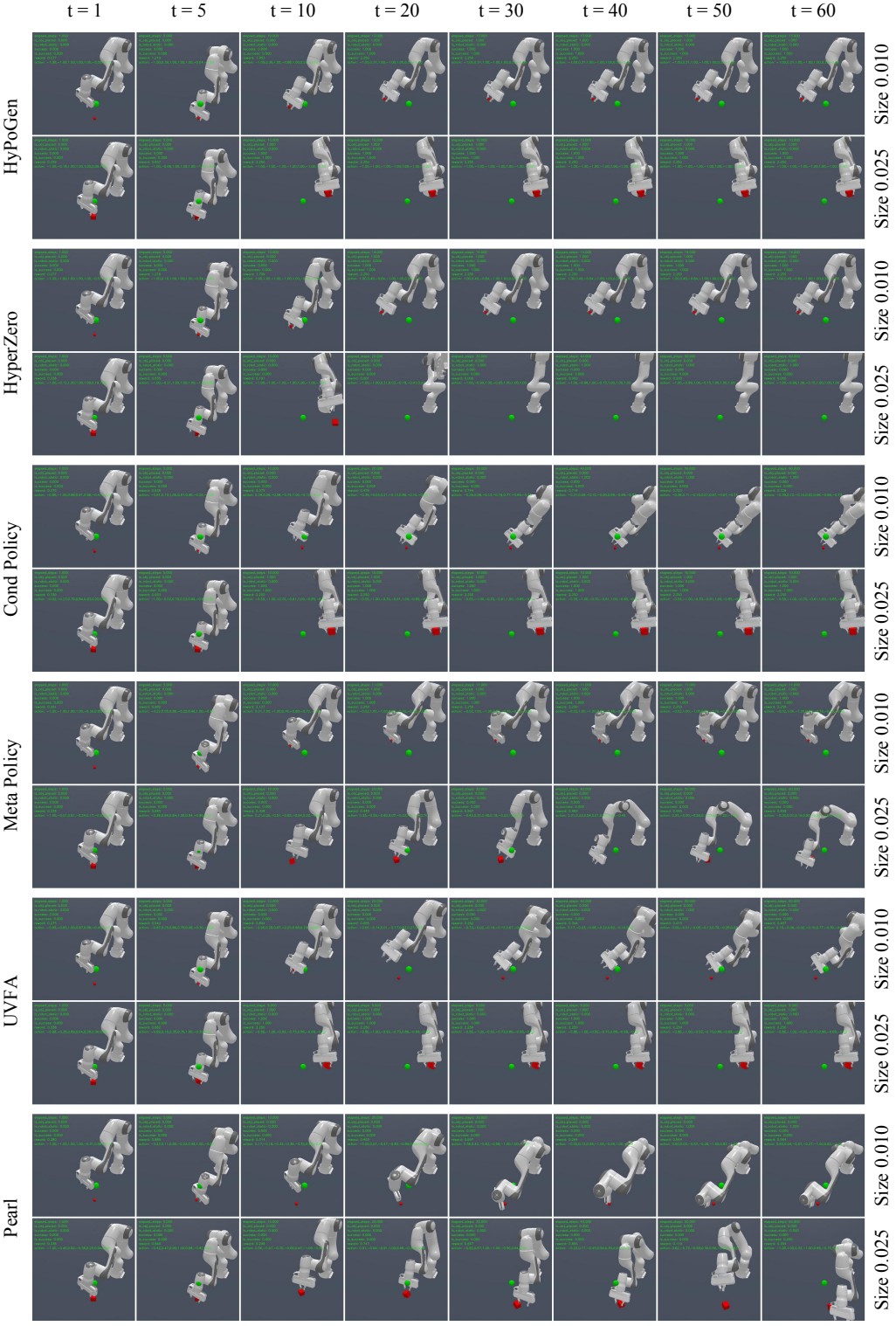

Figure 16: Comparison in LiftCube with different cube sizes.

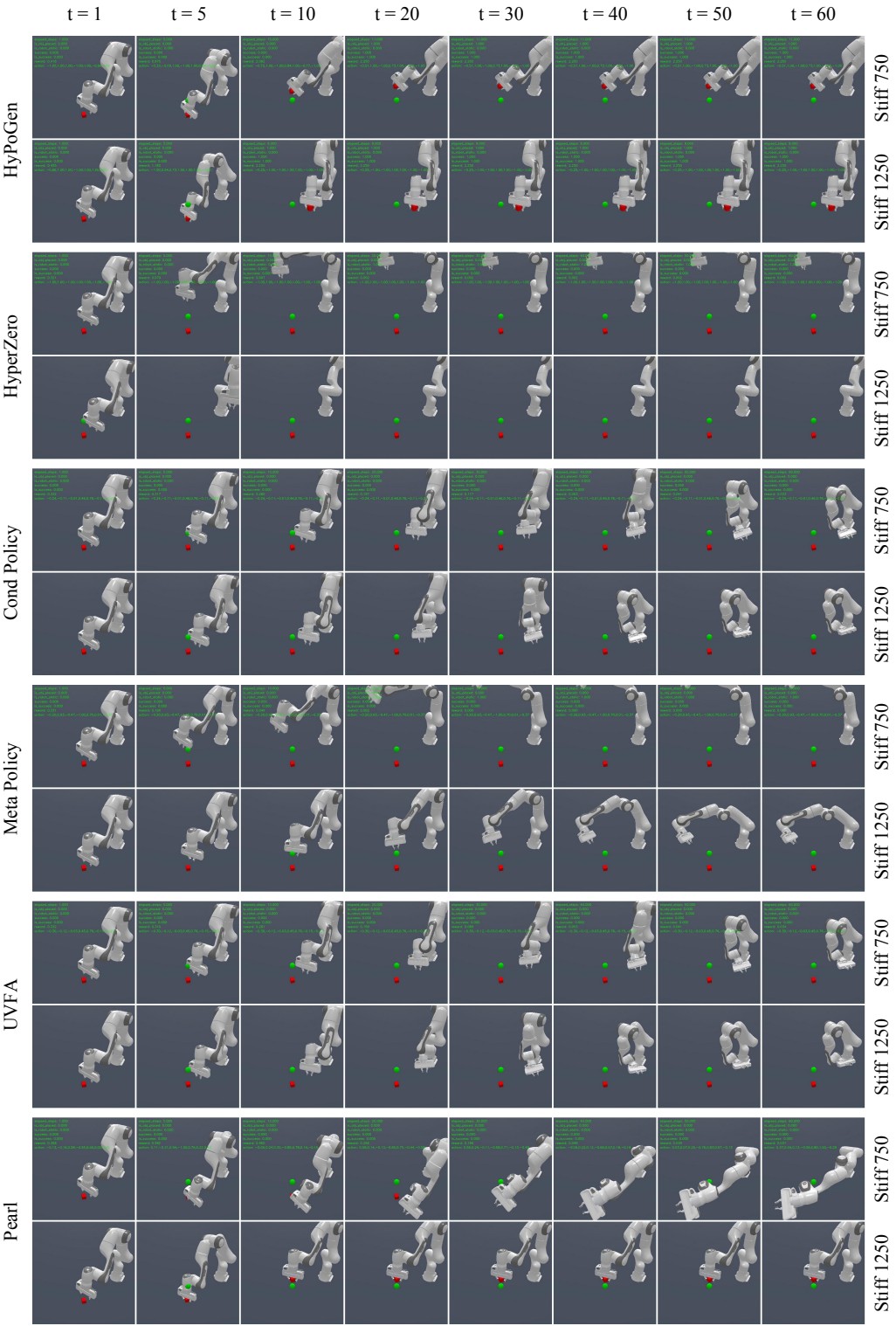

Figure 17: Comparison in LiftCube with different controller stiffness.

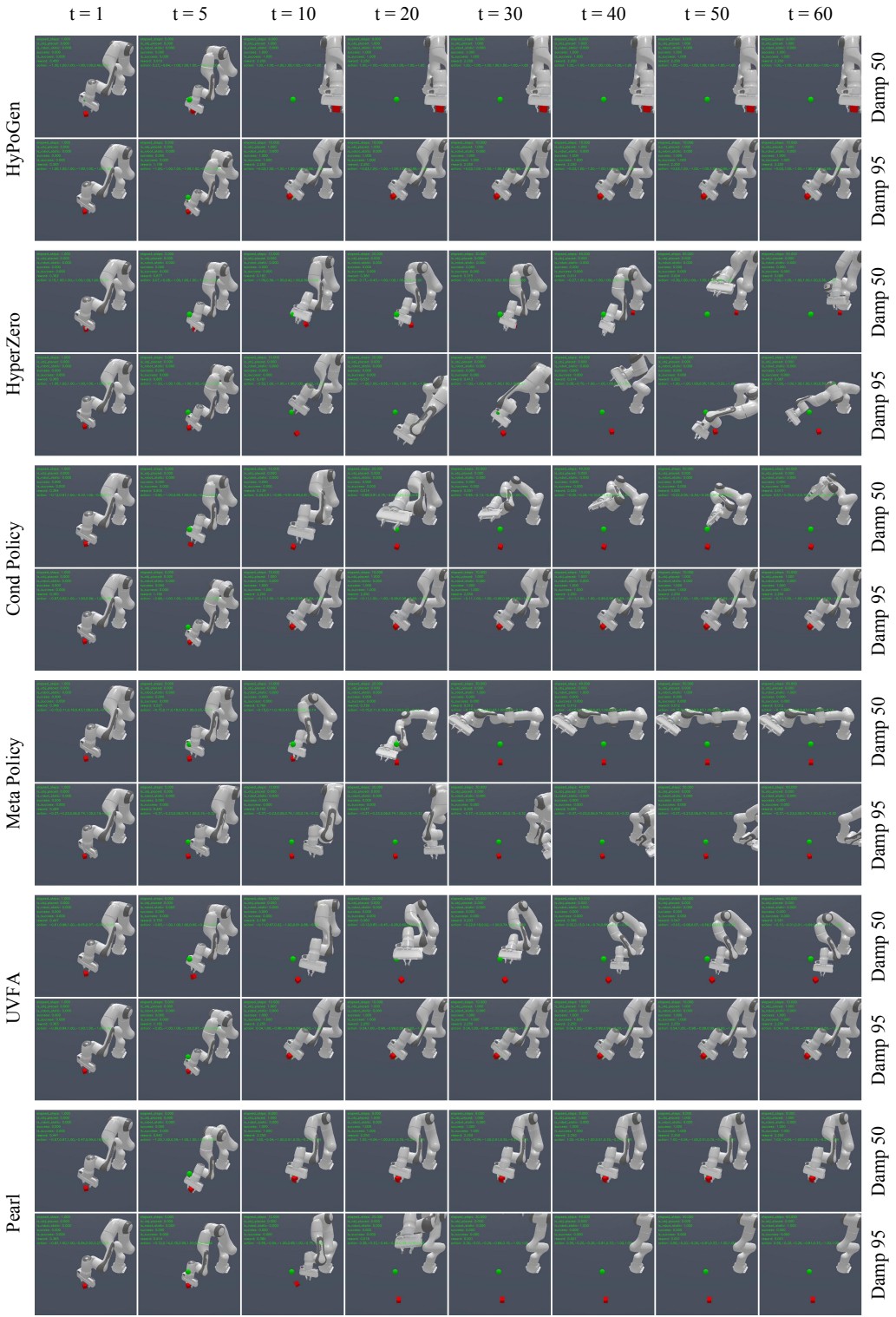

Figure 18: Comparison in LiftCube with different controller damping.

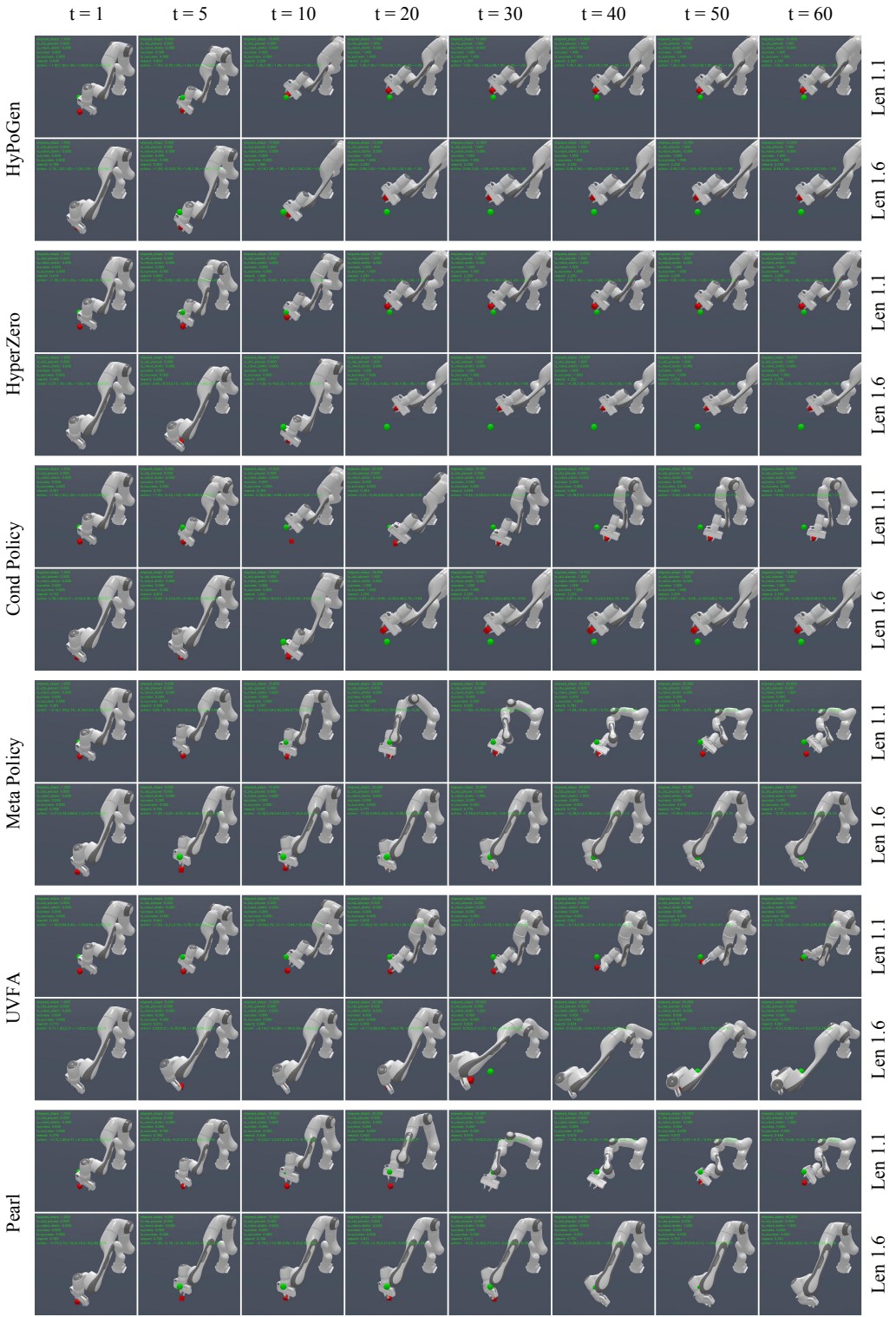

Figure 19: Comparison in LiftCube with different arm lengths.

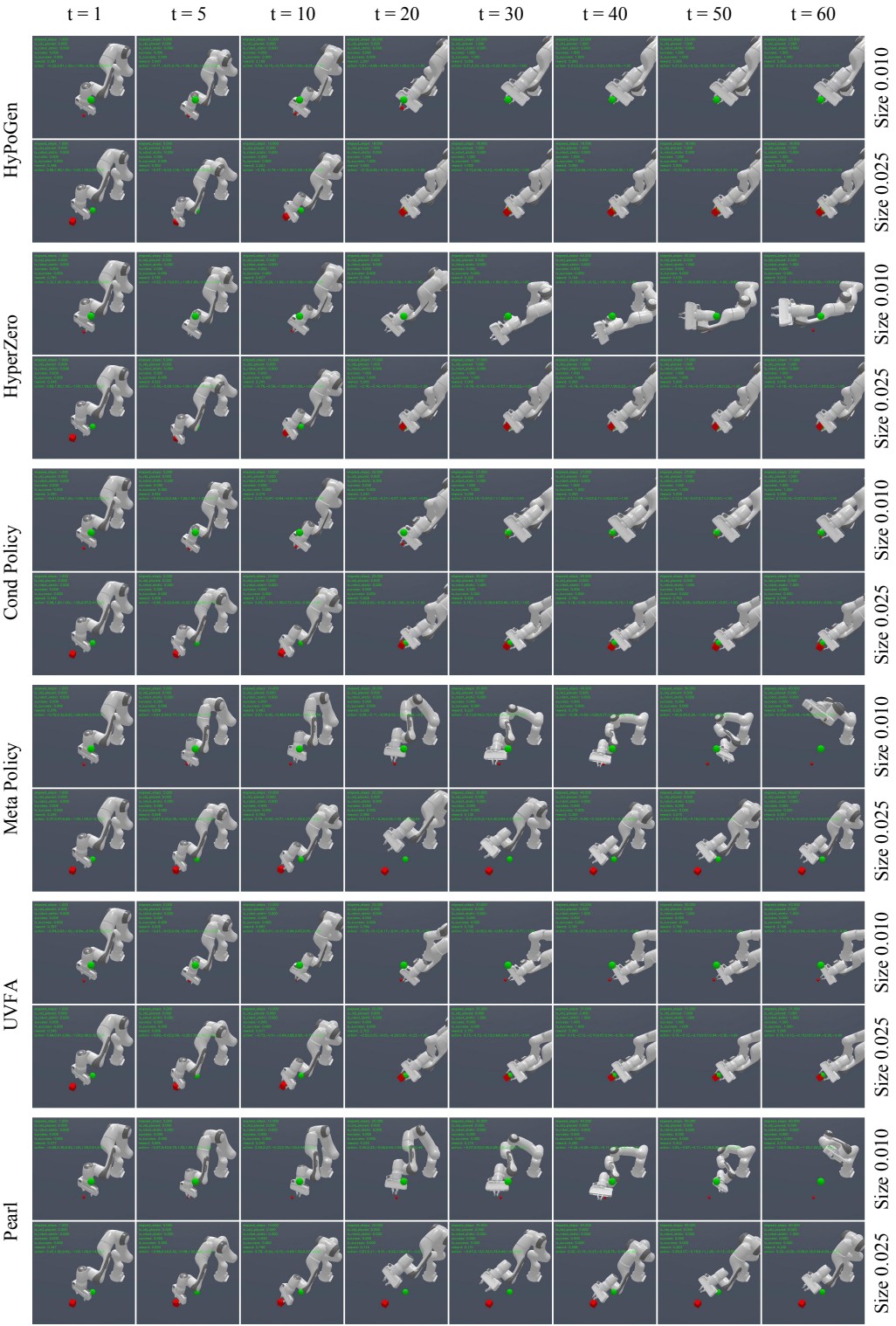

Figure 20: Comparison in PickCube with different cube sizes.

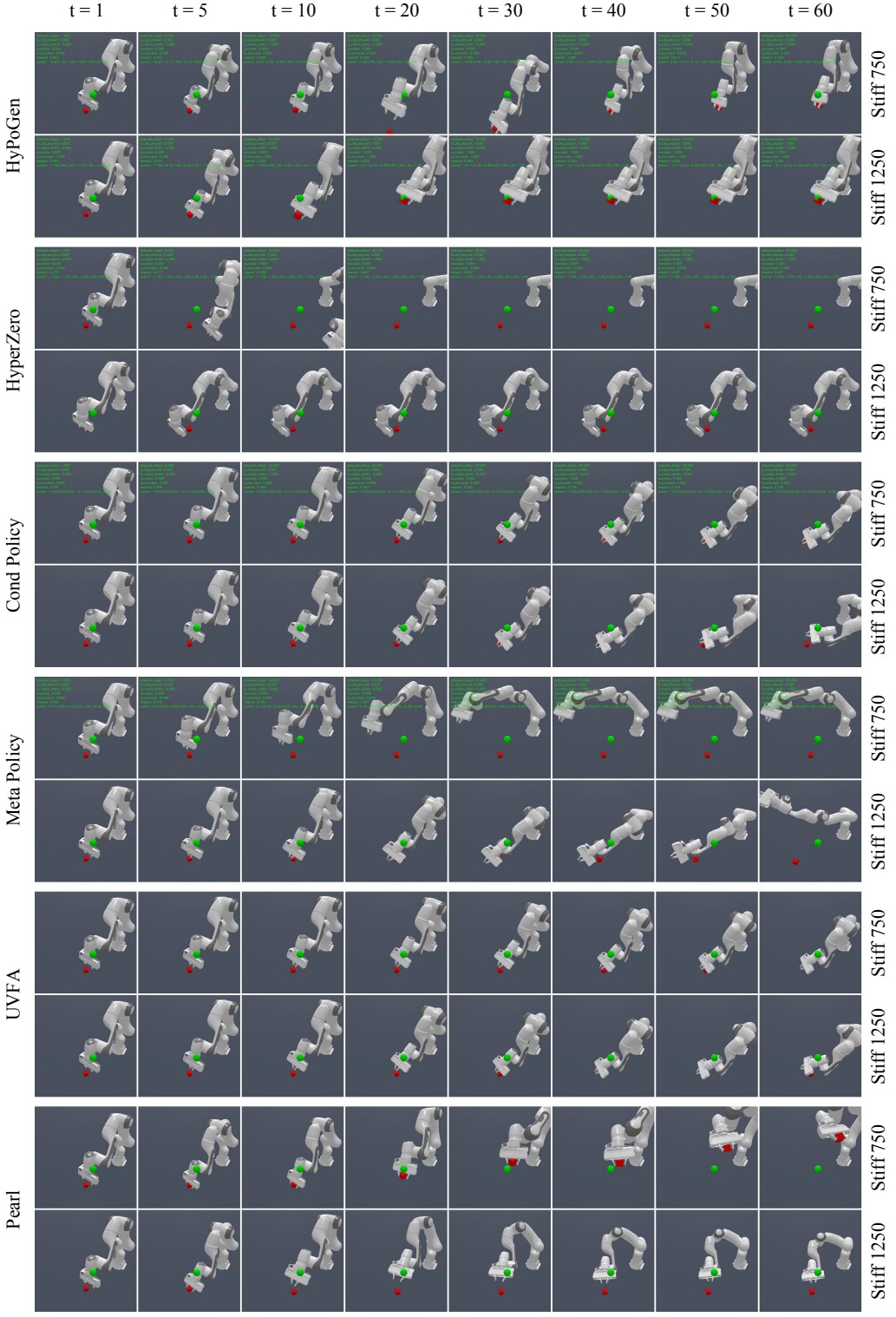

Figure 21: Comparison in PickCube with different controller stiffness.

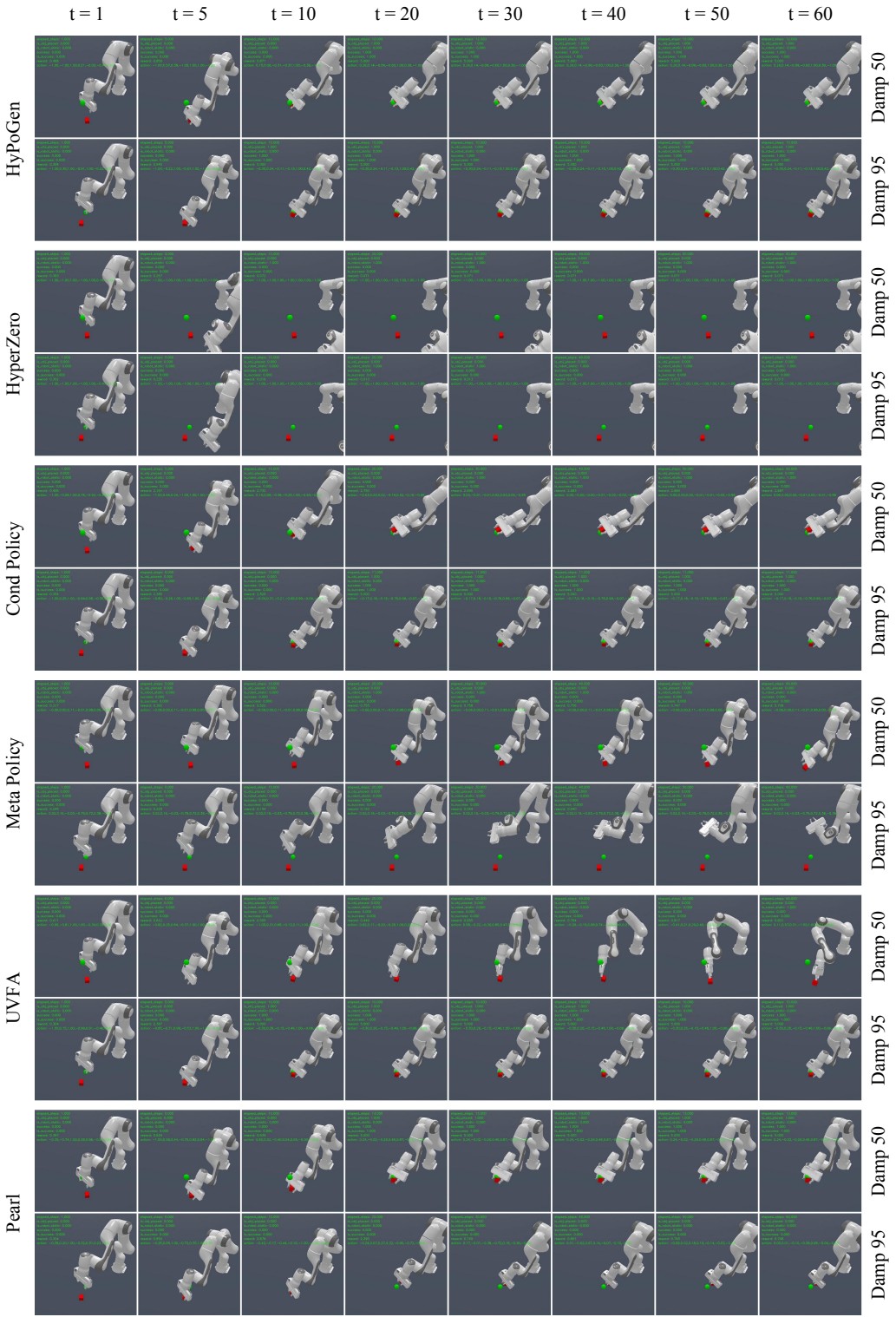

Figure 22: Comparison in PickCube with different controller damping.

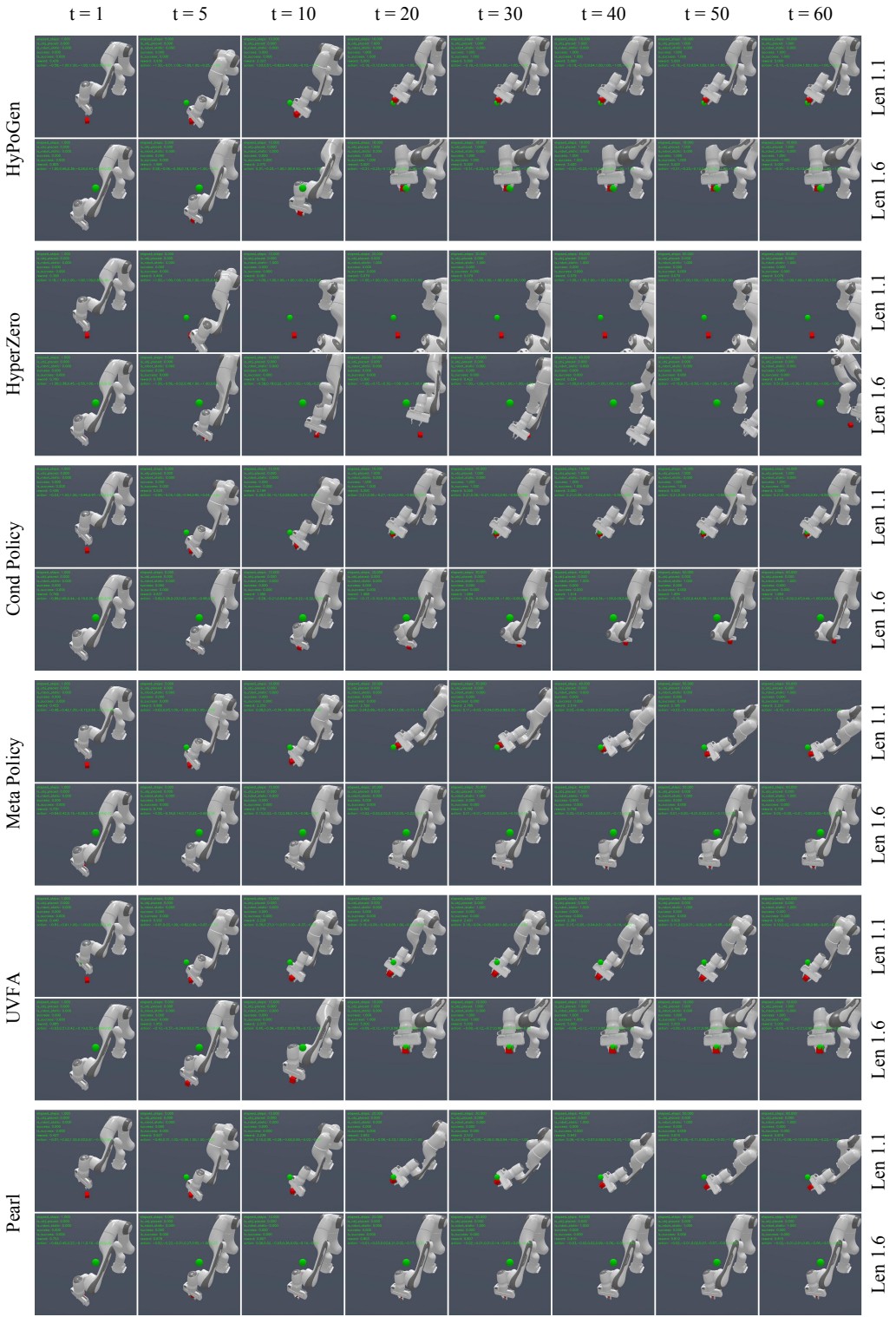

Figure 23: Comparison in PickCube with different arm lengths.

