# OpenReview forum: "HyPoGen: Optimization-Biased Hypernetworks for Generalizable Policy Generation"
_ICLR.cc/2025/Conference — ICLR 2025 Poster_

### Official Review · Reviewer_2Bw1 · 2024-10-29

**Soundness:** 2
**Presentation:** 4
**Contribution:** 3
**Rating:** 6
**Confidence:** 4

**Summary:**

The authors propose to incorporate inductive biases about optimization into a hyper-network framework that generates policies for different tasks. They do so by parameterizing the hypernetwork updates based on the chain rule and also on a delta to the current weights. They show that these inductive biases help HyPoGen outperform previous methods when generalizing across a set of related tasks.

**Strengths:**

- The method is presented and motivated well.
- The experiments provide compelling evidence that HyPoGen outperforms previous methods (at least based on the average).
- The idea of neural gradients for policy generation is interesting. The methods presented in this paper for incorporating optimization inductive biases into hypernetworks could lead to future interesting work.

**Weaknesses:**

- The variances for Table 1 / 2 (reported in Table 13/14) are quite high (on the order of 100-200). This makes it harder to argue that HypoGen is better than the previous methods when the gap is often on the order of a few tens of units. It would be fairer to report exact confidence intervals in the main text to make the comparison more clear.
- The computational costs of HyPoGen aren’t directly addressed. Since it requires multiple steps to keep decreasing the loss, how does the computational cost compare to other methods (especially something like HyperZero)? This makes me think that other baselines could be relevant to assess the tradeoff between computational cost and performance. For example, instead of parametrizing the full chain rule (equation 6/7), can the majority of the benefit be obtained by framing the problem just as a delta from the previous step (equation 5).
- While the paper claims that HypoGen leads to better generalization, Fig. 4,5,6,7,8,9 show generalization somewhat qualitatively and it is not super clear (other than Fig. 4) that HyPoGen generalizes better than the other methods. Are their more precise metrics that could show this generalization more clearly (i.e. report performance within one std of training distribution and outside of that for the relevant methods).

The authors present an interesting method, but I think clearer evaluations and details on the computational requirements would make the paper stronger and help isolate the benefit of the optimization inductive bias.

**Questions:**

- What happens if HyPoGen is applied for many more steps (Fig. D)? Does the loss keep going down, or does it “overfit” eventually?
- What is the strategy for choosing the number of hyper-optimization steps? Do harder problems (more out-of-distribution for example), require more optimization steps?
- Does HyPoGen work for larger policy models / harder environments (the policy networks at the moment seem to be only 2 layer MLPs)?

---

> ### Author Response · Authors · 2024-11-21
>
> Dear Reviewer **2Bw1:**
>
> Thank you for the detailed feedback.  We are pleased to hear that you found the method well-presented and motivated, the experimental results compelling, and the concept of neural gradients for policy generation interesting. We will address your inquiries and concerns on variance analysis, computational trade-offs, and generalization metrics point by point in the following responses.
>
>
>
> **W1:** Thank you for pointing this out, the high variance in our results is due to the inclusion of failed episodes in the calculations, which is an established practice in previous work such as HyperZero. These failed episodes typically have low rewards in the MuJoCo environments or high episode lengths in the ManiSkill environments (around 200 episodes compared to the usual 10–50 for successful episodes), which significantly increase the overall variance.
>
> To address this, we have updated our results to report the standard deviations based only on success episodes. Please refer to Table 24 and 25 in the updated manuscript.
>
> Furthermore, we calculated the probability that HypoGen surpasses all other baselines using formulas from[ this resource](https://stats.stackexchange.com/questions/359768/probability-of-a-difference-between-two-sampling-means-of-two-populations), based on the sample sizes of 50 for MuJoCo and 100 for ManiSkill. These probabilities, presented separately for MuJoCo and ManiSkill, clearly show that HypoGen outperforms all baselines in most cases.
>
> We hope this clarification addresses your concern regarding the comparison with previous methods.
>
> |                   | cheetah-speed | cheetah- length | cheetah- speed&length | finger- speed | finger-length | finger-speed&length | walker-speed | walker-length | walker-speed&length |
> | ----------------- | ------------- | --------------- | --------------------- | ------------- | ------------- | ------------------- | ------------ | ------------- | ------------------- |
> | P(HyPoGen is top) | 1.000         | 1.000           | 1.000                 | 1.000         | 1.000         | 1.000               | 1.000        | 1.000         | 1.000               |
>
> | Method            | LiftCube-cube | LiftCube-stiff | LiftCube-damp | LiftCube-length | Pick&Place-cube | Pick&Place-stiff | Pick&Place-damp | Pick&Place-length |
> | ----------------- | ------------- | -------------- | ------------- | --------------- | --------------- | ---------------- | --------------- | ----------------- |
> | P(HyPoGen is top) | 0.086         | 1.000          | 1.000         | 0.946           | 0.719           | 0.999            | 0.855           | 0.441             |
>
> **W2:** Thank you for raising the important point about computational costs and the trade-off between performance and efficiency. Below, we provide a detailed comparison of computational time between HyPoGen and HyperZero, along with additional experiments to analyze the benefits of parametrizing the full chain rule.
>
> The exact computation costs of the proposed methods and HyperZero are listed below,
>
> |           | WeightGen Time | Rollout Time x 1000 steps |
> | --------- | -------------- | ------------------------- |
> | HyperZero | 0.3ms          | 66.95ms                   |
> | HyPoGen   | 11.62ms        | 66.95ms                   |
>
> While the proposed HyPoGen is slower in the weight generation process, this process only accounts for a small fraction of the total computation time, as the majority of the time is spent on rollout trajectories, which are identical across methods. Furthermore, the policy networks generated by HyPoGen can be reused across multiple trajectory rollouts, making the additional weight generation time non-critical in practice.
>
> To address your suggestion about simplifying the hypernetwork by replacing the full-chain rule modeling (Equation 6/7) with a black-box approach (Equation 5), such as a simple MLP, we conducted experiments on the cheetah task. The results are summarized below:
>
> | Method                          | Rewards        | WeightGen Time |
> | ------------------------------- | -------------- | -------------- |
> | HyPoGen with MLP Hypernet block | 787.33 ± 87.43 | 6.99ms         |
> | HyPoGen (x3 updates)            | 775.55 ± 68.26 | 4.97ms         |
> | HyPoGen (x8 updates)            | 856.88 ± 61.73 | 11.62ms        |
>
> These results demonstrate that replacing the proposed hypernetwork with a simple MLP improves performance over HyperZero, indicating the utility of treating the update as a delta from the previous step. However, explicitly modeling the full chain rule with the proposed hypernetwork leads to further improvements
>
> In terms of computational cost, the black-box hypernetwork achieves performance comparable to a three-layer HyPoGen but is slower in weight generation. This observation underscores the efficiency of parametrizing the full chain rule.
>
> We hope this analysis addresses your concerns and provides clarity on the trade-offs involved.

---

> ### Author Response · Authors · 2024-11-21
>
> **W3:** Thank you for your feedback on the generalization claims and figures. The plotting format in Figures 5–9 follows the conventions of previous works like HyperZero. However, we understand your concern that these qualitative figures may not clearly demonstrate generalization performance.
>
> To better address this, we report the mean rewards for MuJoCo tasks, as well as the success rate and episode length for ManiSkill tasks, under the suggested two conditions: **"within one standard deviation of the training distribution"** and **"outside one standard deviation of the training distribution"**. These metrics are now presented in Table 26, 27, 28, and 29 of the revised manuscript.
>
> The updated results show that HyPoGen outperforms all relevant methods in both in-distribution and out-of-distribution settings across most tasks. This provides stronger evidence of the proposed method's superior generalization ability compared to existing baselines. We hope these additional results clarify our claims and provide a more robust demonstration of HyPoGen’s generalization capabilities.
>
>
>
> **Q1:** We conducted experiments with varying numbers of optimization steps K, with the results presented in Table 20 in the supplementary material. On the cheetah task, the optimal value for K is 8. As K increases beyond this point, the final rewards decrease, indicating potential overfitting.
>
>
>
> **Q2:** The number of hyper-optimization steps (K) is chosen based on experimental results shown in Table 20. We choose K = 8, which provides the best performance, and used this value across all tasks. However, this value can be fine-tuned depending on the specific task or dataset.
>
> To evaluate how the optimal number of steps varies in a more out-of-distribution scenario, we conducted additional experiments using only 10% of the training specifications (compared to 20% in the paper). The results, presented in the table below, show that the best performance was achieved with K = 5, which is smaller than the optimal K for tasks with more training data. This suggests that smaller datasets are more prone to overfitting, necessitating fewer optimization steps to mitigate this issue.
>
> | Layer        | 1              | 3              | 5              | 8              | 10             | 20             |
> | ------------ | -------------- | -------------- | -------------- | -------------- | -------------- | -------------- |
> | Reward ± Std | 554.70 ± 31.79 | 527.25 ± 38.95 | 575.05 ± 55.16 | 517.71 ± 22.09 | 511.09 ± 22.93 | 544.41 ± 30.35 |
>
>
>
> **Q3:** Yes, HyPoGen works for larger policy models and harder environments. We experiment with various sizes of policy networks, as detailed in Table 18 of the supplementary material. The best-performing configuration is a 3-layer policy network. As the number of layers and parameters increases, the learning difficulty of the hypernetwork also increases, which results in decreased performance. Nevertheless, HyPoGen consistently outperforms HyperZero in terms of rewards.
>
> We appreciate your insightful feedback. We hope the additional variance analysis, computational cost comparisons, clearer generalization metrics, and experiments on optimization steps and larger models help address your concerns. Please feel free to let us know if you have more questions that can help improve the final rating of our work.

---

> > ### Comment · Reviewer_2Bw1 · 2024-11-23
> >
> > Thank you for the detailed response from the authors. My main points have been addressed, and I raise my score. Nevertheless, I think that results such as the variance should be reported in the main text, as this is standard with many RL works, and makes it easier for the reader.

---

### Official Review · Reviewer_RPpm · 2024-11-03

**Soundness:** 3
**Presentation:** 3
**Contribution:** 3
**Rating:** 8
**Confidence:** 3

**Summary:**

In low-data settings, using a hypernetwork can be promising. However, classical hypernetworks have critical issues, such as overfitting, which can limit their generalization capability. To address this, the authors modify the role of the hypernetwork. Instead of directly predicting the policy weights, they iteratively compute the weights in a manner similar to multi-step gradient descent. This reformulates the optimization problem for the hypernetwork to predict gradients instead of weights directly, which the authors empirically show leads to better performance and generalization on locomotion and manipulation tasks.

**Strengths:**

This work strengthens hypernetworks by altering the training optimization be guided to output intermediate gradient steps, a novel approach for hypernetworks, especially in policy settings.

The writing and figures are clear, and the experimentation is thorough, with a wide range of baselines that use the same policy network architecture and specification encoder. Performance consistently surpasses that of the baselines.

I found the section "Does HyPoGen actually perform optimization?" particularly interesting and insightful that showcases how it is actually like an optimizer.

**Weaknesses:**

Overall, I think this work is strong, and I would be interested in seeing additional analysis on how closely the predicted intermediate gradients align with true gradients that would result from directly optimizing the policy. While the authors have shown that the iterative optimization process effectively reduces the loss over time, it would be valuable to quantify the similarity between these predicted gradients and the actual gradients obtained through standard gradient descent. Such an analysis could fortify the claim that the hypernetwork is indeed guided to predict meaningful gradients, rather than simply memorizing an update path. This comparison would strengthen the case for the hypernetwork's ability to emulate true gradient-based optimization, supporting its effectiveness in generalizing to unseen tasks. While this is not a critical weakness, it would serve as a valuable addition that could make the work even more compelling.

**Questions:**

I think mentioning learned optimizers : https://arxiv.org/pdf/2209.11208 can be relevant to the overall framing of this work.

I also am wondering how well would directly training the hyper network on the true gradients be, such that instead of it resembling back propagation, it is exactly back propagation (assuming no training error) .

---

> ### Author Response · Authors · 2024-11-21
>
> Dear Reviewer **RPpm**:
>
> Thank you for the thoughtful feedback and for recognizing the novelty, clear presentation, and comprehensive experiments of our work. The majority of your comments revolve around the comparison between the neural gradient and the actual gradient, as well as exploring the performance of our model when trained on ground truth gradients. Next, we address these points as follows:
>
> **W1:** We calculated the cosine similarity between the neural gradients and the corresponding ground truth gradients. A value closer to 1 indicates that the two gradients are more similar in direction. The similarity of parameters after each update is shown in the following. Empirically, we found the similarities to be around 0.366. This suggests that while the neural gradients positively correlated in direction with the ground truth gradients, they differ significantly in actual values. This implies that the neural gradients capture the essence of the exact gradients but are far more effective in updating the parameters given that the neural update process has only 8 steps.
>
> | #upd           | 1      | 2      | 3      | 4      | 5      | 6      | 7      | 8      |
> | -------------- | ------ | ------ | ------ | ------ | ------ | ------ | ------ | ------ |
> | cos similarity | 0.3424 | 0.3662 | 0.3939 | 0.3712 | 0.4116 | 0.3566 | 0.3960 | 0.2888 |
>
> **W2:** We have incorporated a discussion of learned optimizers, including the work you mentioned (https://arxiv.org/pdf/2209.11208), in **Line 155, Related Works** section to provide additional context.
>
> Regarding the question about directly training the hypernetwork on true gradients so that it explicitly mimics backpropagation:
>
> We conducted experiments in the cheetah environment to explore this idea. Specifically, we used PyTorch's autograd function to compute the ground truth gradients of θ after each update and supervised the hypernetwork’s output with an L2 loss. The results are summarized in the table below. As shown, the HyPoGen model trained on ground truth gradients performed poorly in the cheetah environment. This aligns with our discussion in Sections 4.2 and 5.4, where we mentioned that the ground truth gradients are highly data-dependent and noisy, making them unsuitable for effective training.
>
> |                                 | Reward |
> | ------------------------------- | ------ |
> | HyPoGen trained on GT gradients | 115.86 |
> | HyPoGen trained end-to-end      | 819.23 |
>
> We appreciate your detailed feedback and believe that these additions enhance the rigor and comprehensiveness of our work. We look forward to your response and further discussions.

---

> > ### Comment · Reviewer_RPpm · 2024-11-21
> >
> > Thank you, my questions have been addressed.

---

### Official Review · Reviewer_JcCr · 2024-11-04

**Soundness:** 3
**Presentation:** 4
**Contribution:** 3
**Rating:** 8
**Confidence:** 3

**Summary:**

This paper proposes a novel hypernetwork architecture for policy generation in behavior cloning, with a focus on generalization to unseen tasks without requiring extra expert demonstrations. Unlike existing methods that apply hypernetworks to generate policy network parameters directly from a task specification, the proposed hypernetwork architecture, HyPoGen, iteratively generates a policy network by simulating gradient-based optimization, where the pseudo "gradients" are conditioned on the task specification. Additionally, the generation of these pseudo "gradients" across different layers of the policy network is structured to follow the chain rule. The primary idea is that the inductive biases introduced by simulating gradient-based optimization can help improve policy generation quality and improve generalization to unseen tasks. Experimental results show that HyPoGen outperforms baseline methods in generating policies for unseen tasks without any demonstration and that the policy performance improves during the simulated optimization guided by the trained hypernetwork.

**Strengths:**

1. The paper addresses the challenge of generalizing policy generation to unseen tasks without any demonstration, a realistic and difficult problem in imitation learning.

2. The main contribution of the paper is the novel hypernetwork architecture, HyPoGen, which introduces inductive biases to generate a policy network by simulating gradient-based optimization. The architecture performs iterative pseudo "gradient"-descent where the pesudo "gradients" are enforced to follow chain rule across different layers of the policy netowrk, an interesting and sensible approach.

3. It is demonstrated through extensive experiments on locomotion and manipulation benchmarks that HyPoGen outperforms baseline methods in generalizing to unseen tasks without any demonstration. Additionally, several empirical cases studies (notably in Table 4) provide evidence that the proposed hypernetwork architecture effectively simulates an iterative optimization process.

4. The paper is well organized and clearly written, with a thorough discussion of related work to place it within existing research. The rationale behind the HyPoGen architecture design is clearly explained, and the construction of the hypernetwork's inner components is presented effectively.

**Weaknesses:**

1. The training process of the proposed hypernetwork is described somewhat briefly in Section 4.2. Although the set of learnable parameters and the loss function (Eq. 1) are discussed, likely implying end-to-end training over source tasks to minimize the BC loss (Eq. 1) while optimizing all learnable parameters simultaneously, this part could be elaborated more explicitly. Including pseudocode in the appendix might be helpful if the training procedure is not this straightforward.

2. In Section 5.4, there is an analysis of the statistics of the generated policy network parameters with different initial values of $\theta_0$ (Table 3). Are other hypernetwork parameters, apart from $\theta_0$, kept fixed across different initial values of $\theta_0$, or are they retrained separately for each initial value of $\theta_0$? In either case, the presented statistics in Table 3 might not be able to sufficiently answer the question whether HyPoGen remembers a fixed set of parameters for each specification. Specifically, if the other hypernetwork parameters are not retrained, observing the statistics of generated policy network parameters may be less meaningful as the hypernetwork is not adapting to the $\theta_0$ value in use; if the hypernetwork is retrained for different initial values of $\theta_0$, it could still be the case that HyPoGen remembers a fixed set of parameters for each specification for a given $\theta_0$.

**Questions:**

1. Is my understanding of the training procedure of the proposed hypernetwork correct, i.e. are all trainable parameters of the hypernetwork, including $\theta_0$, trained simultaneously in an end-to-end fashion?

2. Are any parts of the hypernetwork parameters retrained for each initial value of $\theta_0$ when performing the analysis of Table 3? See Weakness 2 above for further context of this question.

3. Appendix B.1 provides the range, granularity, and number of samples of tasks specifications in different environments used in the experiments. Is task specification sampling done uniformly? Is the source-test task partition also uniform? If so, for the cases where only one task parameter varies, could there be very similar specifications among the source tasks for most test tasks? For example, Table 6 shows that the Cheetah environment has about 40 possible specifications, and there are 40 samples, so nearly every specification appears in either source or target tasks (assuming sampling without replacement).

4. It is mentioned that the optimization is performed in a compressed latent space. What is the dimension of this latent space?

---

> ### Author Response · Authors · 2024-11-21
>
> Dear Reviewer **JcCr**:
>
> Thank you for the constructive feedback.
>
> We are grateful for your recognition of the novelty of our hypernetwork architecture and the effectiveness of our design choices. Your acknowledgment of our comprehensive experiment results and well-structured presentation is also highly appreciated. Next, we provide the required clarifications regarding the training process and data distribution and more detailed explanations in the following.
>
>
>
> **W1 & Q1:** We apologize for the confusion regarding the training process. The training process is indeed end-to-end, and we have made this clearer in **Line 339, Sec. 4** of the revised version.
>
>
>
> **W2 & Q2:** Thank you for raising this valuable question. For your first question, in our experiments, the other hypernetwork parameters were kept fixed and were not retrained when varying the initial value of $\theta_0$. Regarding your follow-up question, we would like to clarify that the learnable initial weight $\theta_0$ is designed to better adapt in conjunction with the trained hypernetworks. Even though the hypernetworks can perform gradient descent with a random initial $\theta_0$ without retraining, the BC loss associated with our learnable $\theta_0$ is significantly lower compared to a randomized $\theta_0$.
>
> To illustrate this, we present the average BC loss after each update for randomized $\theta_0$ and learnable $\theta_0$ for the Cheetah task below.
>
> |                      | 1       | 2       | 3       | 4       | 5       | 6       | 7       | 8       |
> | -------------------- | ------- | ------- | ------- | ------- | ------- | ------- | ------- | ------- |
> | Learnable $\theta_0$ | 71.4909 | 61.1337 | 46.3077 | 40.6435 | 21.948  | 11.2149 | 3.197   | 1.6058  |
> | Rand $\theta_0$      | 82.8256 | 77.7552 | 67.3428 | 62.6156 | 45.8644 | 33.4012 | 21.0709 | 14.5684 |
>
> This result demonstrates that our model achieves optimal performance with the trained learnable $\theta_0$ while also exhibiting the ability to properly update random initial parameters.
>
>
>
> **Q3:**  We apologize for any confusion. The task specification does not involve random sampling but instead uses evenly spaced values across the range. For instance, in the Cheetah task, all the specifications used during training and testing are defined as `np.linspace(-10, -0.5, 20) + np.linspace(0.5, 10, 20)`.
>
> Regarding the source-task partition, we uniformly sample only 20% of the specifications to use as training tasks, with the remaining 80% serving as testing specifications (as detailed in the “Data Collection and Evaluation Protocols” section of the main text). Thus, for the Cheetah task, we used only 8 specifications for training, which is quite sparse compared to the 32 test specifications.
>
> While it is possible for some test tasks to have similar source tasks, this is not the case for the majority of test tasks due to the limited portion of the training sets.
>
>
>
> **Q4:** The latent dimension is 256 across all experiments.
>
> Once again, thanks for your constructive feedback. Please let us know if there are more comments that help improve the quality of our paper.

---

> > ### Comment · Reviewer_JcCr · 2024-11-22
> >
> > Thank you to the authors for the response. My questions have been overall addressed, and the clarification provided has improved my understanding of the results in Section 5.4.

---

### Official Review · Reviewer_49Ju · 2024-11-04

**Soundness:** 3
**Presentation:** 3
**Contribution:** 2
**Rating:** 6
**Confidence:** 4

**Summary:**

The paper introduces HyPoGen, a hypernetwork that generates policies for unseen tasks without demonstrations by learning policy parameters directly from task specifications. By structuring the network to mimic optimization, HyPoGen generalizes effectively despite limited training data. Experiments show it outperforms state-of-the-art methods on locomotion and manipulation tasks, achieving higher success rates in unseen scenarios. The authors will release the code and models.

**Strengths:**

- The paper was well-written.
- The experimental results show that performance improvement is significant.

**Weaknesses:**

- In general, I am curious about the limitation of the proposed method. Please refer to the questions section.

**Questions:**

- Could the proposed approach work in discrete action space?
    - Is there any potential difficulties or benefits of applying this approach for discrete actions?
- In the experiments section, it seems that the generated policy was adapted in the scenario in which task is slightly modified, such as the desired speed of the agent changed. Can authors provide more experiments that policy is generated for more distinct tasks, such as meta-world?
- Though the authors provide average success rate, it would be also important to show the variance of the success rate.

[1] Meta-World: A Benchmark and Evaluation for Multi-Task and Meta Reinforcement Learning, Tianhe Yu, Deirdre Quillen, Zhanpeng He, Ryan Julian, Avnish Narayan, Hayden Shively, Adithya Bellathur, Karol Hausman, Chelsea Finn, Sergey Levine

---

> ### Author Response · Authors · 2024-11-21
>
> Dear Reviewer **49Ju**:
>
> Thanks for your valuable feedback.
>
> We sincerely appreciate your positive remarks on our paper's presentation and the recognition of significant performance improvement of our method. Since the major concerns are about more extended experiments for more diverse settings and additional variance results, we will address your inquiries and concerns point by point in the following responses.
>
>
>
> **Q1:** Thank you for the insightful question. Theoretically, our hypernetwork design can accommodate both discrete and continuous action spaces. This flexibility is achieved by using one-hot encoding to convert discrete action predictions into a continuous format, making the underlying network operations similar for both action types.
>
> Then the main distinction lies in the training objective, specifically the loss function used. For continuous action spaces, as outlined in Eq. 3, we use an L2 loss function. For discrete action spaces, however, we employ CrossEntropy loss. To provide more clarity, we compare the gradients of the two loss functions below:
>
> - **Gradient of CrossEntropy Loss (CE):** $\frac{∂L}{∂z} = σ(z) - y$
> - **Gradient of L2 Loss:** $\frac{∂L}{∂z} = 2 (z - a)$
>
> where $\sigma$ represents the softmax function, and $y$ is the one-hot label of action $a$.
>
> To further support our claim, we conducted experiments on the CartPole task within the MuJoCo environment. In this task, the objective is to maintain the balance of a standing pole by choosing between pushing the cart either left or right. We varied the pole length from 0.3 to 1.2 and compared the performance of our model against HyperZero. The experimental results are presented in the following.
>
> | method    | mean         |
> | --------- | ------------ |
> | HyperZero | 627.49±40.82 |
> | HyPoGen   | 695.48±21.12 |
>
> The results above align with our theoretical analysis, demonstrating that our Hypernet outperforms traditional hypernetworks and effectively handles discrete action spaces without any issue.
>
>
>
> **Q2**: Thank you for pointing this out. While the tasks may appear slightly modified at first glance, they are more difficult than they seem. As shown in Table 2 of the main paper, all baseline methods struggle with the ManiSkill tasks, even though these tasks differ by only a single parameter. Transferring between them proves to be quite challenging.
>
> To further analyze the limitations of our methods, we conducted additional experiments in the Meta-World environment. Specifically, we trained our hypernetworks on three Meta-World tasks related to buttons: "button-press," "button-press-topdown," and "button-press-wall." We then evaluated our method on a novel task, "button-press-topdown-wall." For all these tasks, we used instruction embeddings from a CLIP-text encoder as the condition for the hypernetworks.
>
> The table below reports the average episode return over 250 episodes.
>
> | Method    | Episode Return |
> | --------- | -------------- |
> | HyperZero | 726.61±70.15   |
> | HyPoGen   | 1138.71±325.54 |
>
> From these results, we can observe that policy networks generated by HyPoGen is much better than HyperZero, which demonstrate the generalization capability of our approach to more distinct tasks in a complex environment like Meta-World.
>
>
>
> **Q3**: Thank you for your valuable suggestion. In addition to reporting the average success rate, we now also report the variance of the success rate in the **Table 23** of the revised manuscript. We can observe that the variance of our proposed are comparable with previous methods. This additional information helps to better understand the stability and robustness of our approach.
>
>
>
> To recap, we demonstrated that our approach can effectively handle discrete action spaces, showing superior performance in the CartPole task compared to HyperZero. Additionally, we provided further experiments in the Meta-World environment, demonstrating the generalization capability of HyPoGen to more distinct tasks. Lastly, we included the variance of success rates to highlight the stability and robustness of our approach compared to other methods.
>
> Thank you again for your constructive feedback. We hope these additional experiments and analyses address your concerns and help improve the final rating of our work. Please feel free to let us know if you have any further questions.

---

### Meta-Review · Area_Chair_XHHa · 2024-12-19

**Metareview:**

The paper presents a significant contribution to policy generation through its novel optimization-biased hypernetwork architecture. While reviewers raised concerns about result variances, computational costs, and generalization metrics, the authors provided comprehensive responses with new experiments showing success-only statistics, minimal computational overhead, and strong quantitative generalization results. The authors also conducted thorough ablations examining different architectures and optimization steps, demonstrating the effectiveness of their design choices. With reviewers increasing or maintaining positive scores following the rebuttal, and given the strong experimental validation, this work warrants acceptance.

**Additional Comments On Reviewer Discussion:**

During the rebuttal period, reviewers raised concerns about result variances, computational costs, generalization metrics, and performance on diverse tasks. The authors addressed these with comprehensive responses including new success-only statistics, detailed cost analysis showing minimal overhead, quantitative generalization results, and new experiments on CartPole and Meta-World. The reviewers were satisfied with these responses. Given the thorough validation and clear technical responses, combined with strong performance and practical importance, this work warrants acceptance.

---

### Decision · Program_Chairs · 2025-01-22

Accept (Poster)